# DYNAMIC DRONE-ASSISTED PICKUP AND DELIVERY ROUTING

## ABSTRACT

We investigate the *dynamic drone-assisted pickup and delivery problem* (DAPDP), which concerns real-time, on-demand routing decisions in scenarios where new paired orders arrive stochastically throughout the day. By leveraging a fleet of trucks each equipped with a drone, operators can split tasks between ground vehicles and aerial vehicles, aiming to minimize total travel costs while respecting constraints on time windows, capacity, and drone flight endurance. We propose a deep reinforcement learning (DRL) approach based on deep Q-learning to decide dynamically which newly arrived orders to dispatch and how to integrate drone sorties effectively. Our experiments on a large, real-world-inspired dataset demonstrate substantial performance gains over greedy, random, and lazy dispatch baselines, yielding 10.6%, 22.6%, and 37.2% savings, respectively, in total travel cost. Additionally, our value-based policy learns *subset selection* decisions that co-adapt with a paired sub-solver, yielding near-oracle performance and outperforming classical and PPO baselines.

## 1 INTRODUCTION

The rapid expansion of e-commerce and ever-increasing customer demand for instantaneous deliveries have pushed logistics and supply chain providers to optimize last-mile operations (Pillac et al., 2013; Psaraftis et al., 2016; Voccia et al., 2019). The last mile is widely recognized as the most time-consuming and expensive segment of the supply chain, often accounting for up to 30% of total shipping costs. Consequently, developing innovative methods for fast, cost-effective, and reliable delivery has become an urgent priority (Murray & Chu, 2015).

**Drone-Assisted Delivery.** Unmanned Aerial Vehicles (UAVs), commonly referred to as drones, have gained traction as a complementary tool to ground vehicles (trucks or vans) (Ponza, 2016). While drones may have limited payload capacity and flight range, their potential for bypassing traffic congestion and traveling straight-line distances can yield notable efficiency improvements. They can reduce not only the physical distance traveled by trucks but also the delivery time, particularly for high-priority parcels. However, integrating drones into an existing ground-based delivery system requires careful coordination, as multiple interdependent decisions arise: Where should a drone be launched? Which requests can be feasibly served by a drone? When should the truck wait or proceed, and how should it rendezvous again with the drone?

**Dynamic Nature.** The *dynamic* aspect of this problem imposes additional complexity. In many scenarios, especially in on-demand or same-day delivery, requests are not known fully in advance. Instead, they appear stochastically throughout the day, each with a required time window in which service must occur. Standard static approaches become suboptimal if they ignore newly arrived requests after route dispatch.

In this work, we analyze the *dynamic drone-assisted pickup and delivery problem* (DAPDP), where a fleet of trucks, each with an onboard drone, must make repeated dispatch decisions over multiple epochs (e.g., hourly). We aim to minimize the total cost (or total distance) across all epochs. This cost is composed mainly of the driving distance of trucks plus a smaller drone distance penalty, subject to drone endurance limits, time windows, and capacity constraints.

**Key Challenges.** (i) *Coordination of Vehicles*: The simultaneous control of trucks and drones is a nontrivial scheduling and routing challenge. (ii) *Rolling Horizon Updates*: Requests appear gradually,

requiring repeated re-optimization of routes without significantly disrupting ongoing service. (iii) *Large Combinatorial Action Space*: At each epoch, deciding which requests to dispatch and which to defer grows exponentially with the number of unserved requests. (iv) *Tradeoff Between Immediate and Future Dispatch*: A request might become cheaper to serve if partially consolidated with future arrivals, but deferring requests too long risks losing feasibility or incurring penalties.

**Positioning and novelty.** While we employ a standard DQN, the learning task is specific and non-trivial. Our contributions arise not from introducing a new reinforcement learning (RL) primitive, but from how RL is *coupled* to the drone-assisted routing setting. First, *state design*: the agent observes a drone-aware representation that encodes eligibility, endurance slack, line-of-sight (LoS) to road detour ratio, and local clustering, enabling it to reason explicitly about launch–rendezvous structure and endurance-limited sorties while respecting pickup–delivery precedence and time windows. Second, *per-request Q-targets coupled to a static sub-solver*: the network outputs request-wise Q-values for "dispatch" vs. "defer," and these values are trained using global rewards derived from a fixed, transparent static solver. This isolates the learned component to *dispatch timing and subset selection*, leaving routing and feasibility handled by the sub-solver. Third, *empirical solver–policy co-adaptation at city scale*: the resulting policy is demonstrably drone-aware, i.e., it defers or consolidates drone-suitable work when beneficial and achieves near-oracle performance across real metropolitan geometries. Together, these elements constitute the methodological contribution of the paper: an RL formulation that meaningfully integrates with, rather than replaces, a strong optimization-based sub-solver.

## 2 PROBLEM FORMULATION

This section builds upon the static DAPDP definition and extends it to a dynamic environment. We summarize the critical elements below. The full static DAPDP formulation is provided in Appendix A.

### 2.1 STATIC DAPDP WITH PAIRING

Table 1: Notation used in the static and dynamic DAPDP.

| SYMBOL | MEANING |
|---|---|
| $\mathcal{V}$ | Set of trucks (each carries one drone) |
| $\mathcal{R}$ | Set of pickup–delivery requests, indexed by $r$ |
| $(p_r, d_r)$ | Pickup and delivery nodes for request $r$ |
| $\mathcal{N}$ | Node set (depot 0 plus all pickups/deliveries) |
| $d_{ij}, d_{ij}^{\mathrm{dr}}$ | Road distance (truck) and flight distance (drone) |
| $[e_i, \ell_i]$ | Time window at node $i$; $T_i$ service time |
| $E$ | Drone endurance budget (flight time) |
| $x_{ij}^v \in \{0,1\}$ | Truck $v$ travels $i \to j$ |
| $y_{ijk}^v \in \{0,1\}$ | Drone of truck $v$ flies $i \to j \to k$ (launch $i$, serve $j$, rejoin $k$) |
| $u_i^v$ | Service start time of truck $v$ at node $i$ |
| $w_i^v$ | Load of truck $v$ on arrival at $i$ |
| $z_{r,p}^{\mathrm{dr}}, z_{r,d}^{\mathrm{dr}} \in \{0,1\}$ | Drone serves the pickup or delivery node of request $r$ |
| $\alpha$ | Relative cost weight for drone distance |

Trucks can serve multiple requests directly, while drones can be dispatched for certain requests if feasible (payload, time windows, flight range). Given the notation in Table 1, the objective is $\min \sum_v \sum_{(i,j)} d_{ij} x_{ij}^v + \alpha \sum_v \sum_{(i,j,k)} d_{ij}^{\mathrm{dr}} y_{ijk}^v$, which utilizes travel distance as a proxy for cost. Each request $r \in \mathcal{R}$ has paired nodes $(p_r, d_r)$. In addition to flow, capacity, and time-window constraints (Appendix A), we enforce:

(*visit exactly once*) $\quad \sum_v \sum_j x_{p_r j}^v + \sum_v \sum_{i,k} y_{i p_r k}^v = 1, \quad \sum_v \sum_j x_{d_r j}^v + \sum_v \sum_{i,k} y_{i d_r k}^v = 1, \ \forall r$

$$(1)$$

(*pickup precedes delivery on the carrying truck*)

$$u_{d_r}^v \ \geq \ u_{p_r}^v + T_{p_r} - M\Big(1 - \sum_j x_{p_r j}^v\Big), \ \forall r, v \tag{2}$$

(*one-leg-by-drone at most*) $\quad z_{r,p}^{\mathrm{dr}} + z_{r,d}^{\mathrm{dr}} \leq 1, \ \forall r \tag{3}$

(*link $z^{dr}$ to sorties*) $\quad z_{r,p}^{\mathrm{dr}} \leq \sum_v \sum_{i,k} y_{i \, p_r \, k}^v, \quad z_{r,d}^{\mathrm{dr}} \leq \sum_v \sum_{i,k} y_{i \, d_r \, k}^v \tag{4}$

(*transfer timing if drone serves a leg*)

$$u_k^v \geq u_{p_r}^v + T_{p_r} + d_{p_r k}^{\mathrm{dr}} - M(1 - z_{r,p}^{\mathrm{dr}}), \tag{5}$$

$$u_{d_r}^v \geq u_i^v + d_{i d_r}^{\mathrm{dr}} - M(1 - z_{r,d}^{\mathrm{dr}}) \tag{6}$$

Constraints 3–6 capture the single-visit drone sortie ($i \to j \to k$): if the drone serves $p_r$ ($d_r$), the truck must rendezvous at $k$ (launch at $i$) in time. Endurance constraints apply to each sortie:

$(d_{ij}^{\mathrm{dr}} + d_{jk}^{\mathrm{dr}}) \leq E$. Constraints such as time windows are treated as hard constraints in the static sub-problem. Any dispatch subset that cannot be feasibly routed by the sub-solver (e.g., due to unavoidable time window violations) results in a penalty for the agent, as described in Section 4.1.

## 2.2 DYNAMIC SETTING

In a dynamic scenario, requests $\mathcal{R}$ are partially unknown initially. They arrive in discrete time epochs $t = 1, \ldots, T$. At each epoch:

- New requests $\mathcal{R}_t$ are sampled randomly without replacement from a pool (with known locations, demands, earliest start $e_i$, and latest start $l_i$).
- The system chooses a subset $S_t \subseteq \mathcal{R}_{\mathrm{active}}$ to dispatch, solving a static sub-problem restricted to those requests.
- The cost $C(S_t)$ is accrued, and the agent observes a reward $r_t = -C(S_t)$.
- The environment transitions to $t + 1$, revealing newly arrived requests $\mathcal{R}_{t+1}$. Requests not served remain active if their windows still permit future service.

A request becomes a *must-go* if $l_i$ is approaching such that deferral is no longer feasible. Requests are allowed to be deferred within the horizon, but by the final epoch $T$, they must be served.

**Goal.** Maximize cumulative reward (minimize total travel cost) over $T$ epochs, subject to feasibility each step.

## 3 APPROACH: DEEP REINFORCEMENT LEARNING (DRL)

We model the dynamic DAPDP as a Markov Decision Process (MDP) and solve it with Deep Q-learning. Below, we outline each MDP component and the associated neural network architecture. Further details of our approach appear in Appendix C.

### 3.1 MDP COMPONENTS

**State $s_t$.** At epoch $t$, the MDP state is

$$s_t = \big(g_t, \{x_{t,r}\}_{r \in R_t^{\mathrm{active}}}\big), \tag{7}$$

where $g_t \in \mathbb{R}^{d_g}$ collects global features and $x_{t,r} \in \mathbb{R}^{d_x}$ is the feature vector of active request $r$. **Global features:** epoch index $t$, time remaining, number of active requests $|\mathcal{R}_{\mathrm{active}}|$, *fleet-side drone availability* (count of drones not committed in the current sub-solution), *weather/airspace proxy*. **Per-request features:** pickup and delivery coordinates, demands, time windows, service times, a must-go flag, and $k$-NN distances. Per-request features also augment core attributes with drone-aware signals: (i) *Eligibility and payload*: drone-eligibility indicator, payload estimate. (ii) *Endurance slack*: maximum feasible sortie time budget for each request given drone endurance and speed, and the nearest feasible launch/rejoin pair. (iii) *LoS/road detour ratio*: $\frac{d_{\mathrm{LoS}}^{\mathrm{dr}}(i,j)}{d_{\mathrm{road}}(i,j)}$, which is the ratio of straight-line (drone) distance to road-network (truck) distance, serving as a proxy for potential drone advantage. (iv) *Local opportunity*: $k$-NN (we use $k=5$) density of *drone-eligible* neighbors within a fixed radius, for consolidation potential. (v) *Window slack flags*: indicators for tight/mid/loose time-window slack. Removing endurance slack or LoS/road reduces performance by 2.2% and 1.6%, respectively (Table 11), confirming that the agent leverages drone affordances rather than learning a drone-agnostic VRP heuristic. Details of state encoding in Appendix C.1.

**Action $a_t$.** Specifies a binary decision (dispatch or defer) for each active request. At each epoch, the dispatched subset is passed to a static DAPDP sub-solver (see Section 4.1 / Appendix E.2), which returns routes and cost under a per-epoch time limit.

**Reward $r_t$** Let $C_t = \mathrm{TruckDistance}_t + \alpha\,\mathrm{DroneDistance}_t$ denote the routed cost for the subset $S_t$ dispatched at epoch $t$, and let $\Delta C_t = \max\{0, C_{t-1} - C_t\}$ with $C_0 := 0$. We define the global reward as

$$r_t = \begin{cases} -C_t + \lambda_{\mathrm{shape}}\,\Delta C_t, & \text{if the sub-solver returns a feasible solution,} \\ r_{\mathrm{penalty}}(t), & \text{if the action leads to an invalid solution,} \end{cases} \tag{8}$$

where

$$r_{\text{penalty}}(t) = -\beta \left( c_{\text{base}} + \sum_i v_i(t) \right),\tag{9}$$

where $\beta = 2$, $c_{\text{base}}$ is the 95th percentile of feasible-solution costs over a rolling window of 256 episodes, and $v_i(t)$ is the magnitude of violation of constraint $i$ at epoch $t$. We then assign a per-request reward

$$r_t^{(i)} = \begin{cases} r_t/|S_t|, & \text{if request } i \text{ was dispatched at epoch } t \text{ and } |S_t| > 0, \\ 0, & \text{otherwise,} \end{cases}\tag{10}$$

which is used in the Q-learning target for each request (Section 3.2.2). Maximizing the cumulative return thus corresponds to minimizing the sum of routed distances, while discouraging infeasible actions through $r_{\text{penalty}}(t)$ (see Section C.2.3 for details).

**Environment** We wrap a time-limited static DAPDP solver as an RL environment with `environ_reset` and `environ_step`. At each epoch, the *action* is a binary dispatch/defer vector over active requests; the environment routes the dispatched set with the sub-solver and returns the next *observation* (global + per-request features), a *reward*, and a *done* flag. An *episode* spans all epochs of one dynamic instance, so maximizing return is equivalent to minimizing the sum of per-epoch route costs. Environment implementation details in Appendix C.2.

**Transition.** The environment checks feasibility, calculates cost, applies a large negative penalty for infeasibility, and reveals new requests for $t + 1$.

**Termination** If either the final epoch is reached or the agent submits an invalid solution, the environment reaches a terminal state, and a substantial penalty (Equation (9)) is imposed as feedback to the agent through the reward.

## 3.2 Deep Q-Network Method

We represent $Q(s, a; \theta)$ by a multi-layer feedforward neural network with parameters $\theta$. We adopt the classic Q-learning update (Sutton & Barto, 2018):

$$Q(s_t, a_t) \leftarrow Q(s_t, a_t) + \alpha^l \left[ r_t + \gamma \max_{a'} Q(s_{t+1}, a') - Q(s_t, a_t) \right].$$

In practice, we use a *target network*, experience replay, and mini-batch stochastic gradient descent to stabilize training (Mnih et al., 2015).

### 3.2.1 Network Architecture

**Input Layer** The state $s_t$ provided to the DQN comprises features encoding the dynamic environment. The collection of global features and the set of $\mathcal{R}_{\text{active}}$ per-request feature vectors form the input to the DQN. Since the number of active requests $\mathcal{R}_{\text{active}}$ varies, padding is employed to ensure a consistent input dimension for the network.

**Hidden Layers** The DQN employs a multi-layer perceptron (MLP) architecture with sizes [128, 64, 32, 16, 8] to process these features.

**Output Layer and Per-Request Action Scores** For each of the $\mathcal{R}_{\text{active}}$ active requests currently being considered, the network outputs two Q-values (scores): $Q(s_t, \text{dispatch}_i; \theta)$ representing the expected return if request $i$ is chosen for dispatch in the current epoch, and $Q(s_t, \text{defer}_i; \theta)$ representing the expected return if request $i$ is deferred.

**Forming the Dispatch Subset $S_t$ (Action Selection)** Based on these Q-values, a decision is made for each active request $i \in \{1, \ldots, |\mathcal{R}_{\text{active}}|\}$. Specifically, an $\epsilon$-greedy policy selects between dispatching request $i$ or deferring it:

- With probability $1 - \epsilon$, action $\mathcal{R}^*_{\text{active, i}} = \arg\max_{\mathcal{R}_{\text{active, i}} \in \{\text{dispatch}_i, \text{defer}_i\}} Q(s_t, \mathcal{R}_{\text{active, i}}; \theta)$ is chosen.
- With probability $\epsilon$, a random action (dispatch or defer) is chosen for request $i$.

The overall action $a_t$ for the epoch is the set of these $\mathcal{R}_{\text{active}}$ individual decisions. The subset of requests $S_t = \{\text{request } i \mid \text{the chosen action for } i \text{ was dispatch}_i\}$ is then passed to the static DAPDP sub-solver.

### 3.2.2 Q-VALUE ESTIMATION AND TRAINING UPDATE

After the dispatch subset $S_t$ is determined by making decisions for all requests in the set of active requests $\mathcal{R}_{\text{active}}$, the sub-solver computes the actual cost $C(S_t)$ incurred to service these specific requests. The immediate global reward for the epoch is $r_t = -C(S_t)$ (this may also include penalties for infeasibility or additions from reward shaping, as per Section 4.2).

The Q-network's parameters $\theta$ are updated using mini-batches sampled from an experience replay buffer. For each transition $(s_t, a_t, r_t, s_{t+1})$ in a batch, where $a_t$ represents the collection of individual dispatch/defer decisions for all requests in $\mathcal{R}_{\text{active}}$:

For each request $i \in \mathcal{R}_{\text{active}}$, let $d_i^{\text{chosen}}$ be the specific action (either dispatch$_i$ or defer$_i$) taken for request $i$ as part of the composite action $a_t$ at state $s_t$. The target value $y_i$ for updating $Q(s_t, d_i^{\text{chosen}}; \theta)$ is computed as:

$$y_i = \text{reward\_for\_decision}_i + \gamma \max_{d_i' \in \{\text{dispatch}_i, \text{defer}_i\}} Q(s_{t+1}, d_i'; \theta^-)$$

where $\theta^-$ are the parameters of the target network, and $\gamma$ is the discount factor.

The component reward_for_decision$_i$ is defined based on the chosen action $d_i^{\text{chosen}}$ and the global reward $r_t$:

$$\text{reward\_for\_decision}_i = \begin{cases} r_t/|S_t| & \text{if } d_i^{\text{chosen}} = \text{dispatch}_i, \ |S_t| > 0 \\ 0 & \text{otherwise} \end{cases}$$

Exact marginal contributions would require solving multiple counterfactual static DAPDP instances per epoch, which is computationally prohibitive. The loss for updating the Q-network is then typically calculated as the sum of squared differences between the target $y_i$ and the predicted Q-value $Q(s_t, d_i^{\text{chosen}}; \theta)$, averaged over all individual request decisions in the mini-batch:

$$\mathcal{L}(\theta) = \mathbb{E}_{(s_t, a_t, r_t, s_{t+1}) \sim \mathcal{B}} \left[ \sum_{i \in \mathcal{R}_{\text{active}}} (y_i - Q(s_t, d_i^{\text{chosen}}; \theta))^2 \right]$$

where $\mathcal{B}$ is the replay buffer, and the sum is over all requests $i$ in the set of active requests $\mathcal{R}_{\text{active}}$ for state $s_t$.

## 4 EXPERIMENTS

To assess the effectiveness of our proposed DRL approach, we conducted a comprehensive set of experiments on large-scale, real-world-inspired data. This section details the experimental setup, baselines, main performance results, ablation studies, and hyperparameter sensitivity analyses. The environment is implemented in Python 3.10, with PyTorch for the DQN and a custom Python-based simulator for request sampling and state construction (details in Appendices E.1 and C.2, respectively). In deployment, each decision epoch requires a single forward pass of the Q-network ($< 10$ ms on a V100 GPU) and one call to the ALNS static sub-solver (2–5 s for $\leq 100$ active requests).

### 4.1 EXPERIMENTAL SETUP

**Data provenance.** We use 300 anonymized truck-only delivery days from a large urban courier (two metro areas; UTM projection), each with 100–200 requests, time windows, and service times. Dynamic instances are created by sampling without replacement over 5–9 epochs (Appendix C.2.2). Our use case does not require a drone-native operator; we overlay drone feasibility on anonymized truck-only logs, so the relevant stochasticity is the order distribution and time-window structure, not prior drone operations.

**Truck and Drone Parameters.** We assume a sufficiently large pool of homogeneous trucks is available, each with a capacity of 200 parcels and each equipped with a drone subject to an endurance

of $E$ (15-25 minutes). We set the relative cost of the drone to truck $\alpha$ to 0.3. A request is drone-eligible if the payload $\leq 2.3$ kg and a feasible sortie $(i \rightarrow j \rightarrow k)$ exists with $(d_{ij}^{\mathrm{dr}} + d_{jk}^{\mathrm{dr}})/v_{\mathrm{dr}} \leq E$. Unless otherwise noted, we set drone speed $v_{\mathrm{dr}} = 29$ m/s ($\sim$65 mph), range 10–12 miles round-trip, and payload $\leq 2.3$ kg in line with minimum public specifications of Wing and Zipline's P2 systems (Amazon MK30 payload limit 5 lb). We report sensitivity to $(E, v_{\mathrm{dr}}, \text{payload})$ in Section 4.6.

When the DRL agent selects a subset of orders $S_t$ for dispatch, the static DAPDP sub-solver determines the routes and the minimum number of truck-drone pairs from this pool required to serve $S_t$, respecting all capacity and endurance constraints. The objective function (Section 2.1), by minimizing total travel distance, implicitly encourages the use of fewer vehicles. This 'effectively unrestricted' fleet assumption ensures any feasible subset of orders can be routed, allowing focus on the dynamic dispatch decision. The impact of finite fleet capacity is explored in our generalization experiments (Appendix D.8).

**Baselines.** We include two classical and one RL baseline that share the same sub-solver and compute budget per epoch as our DQN inference+solve:

1. **Greedy Dispatch**: Dispatch all newly arrived requests immediately.
2. **Random Dispatch**: Choose a random fraction of new requests to dispatch each epoch.
3. **Lazy Dispatch**: Only dispatch requests that must be served (time window closing).
4. **Oracle**: We add a clairvoyant *oracle* that assumes full future knowledge. For each *full* day (5–9 epochs), the oracle is given *all* requests that will arrive across the entire horizon before the first decision. It then solves a single static DAPDP that respects all time windows and endurance constraints (using the same static DAPDP sub-solver but with a 60-second time limit; details in Appendix E.2) and reports the day's minimum total cost. This cost is a hindsight lower bound for any online policy.
5. **MSA/SAA (scenario-based planning)** We implement a rolling-horizon Multiple Scenario Approach (Bent & Van Hentenryck, 2004): for each epoch, sample $K \in \{10, 30\}$ demand scenarios for future epochs, solve each static realization, and dispatch by consensus.
6. **Waiting/relocation** Following Mitrović-Minić & Laporte (2004), we allocate route slack to must-go zones and delay non-urgent requests.
7. **PPO (actor–critic)** We implement Proximal Policy Optimization with a 64–64 MLP actor–critic, discrete per-request actions, and the same wall-clock training budget as DQN.

INFINITE-FLEET ASSUMPTION & INVALID SOLUTION HANDLING

To guarantee feasibility, we follow industry practice and assume an unrestricted vehicle fleet. While theoretically any request can thus be served, the objective still penalizes superfluous trips, incentivizing minimal fleet use. When the agent proposes an infeasible action (e.g., serving after a time window expires), the episode terminates and a structured penalty $r_{\mathrm{penalty}} = -\beta(c_{\mathrm{base}} + \sum_i v_i)$ with $\beta{=}2$ is applied, where $c_{\mathrm{base}}$ tracks the worst valid cost seen, and $v_i$ is the magnitude of the violation of constraint $i$. Quantitative robustness experiments with respect to $\beta$ are reported in Table 9.

STATIC DAPDP SUB-SOLVER (ALNS WITH PAIRING)

We implement an *adaptive large neighborhood search* (ALNS) algorithm adapted to our paired pickup–delivery with single-leg drone sorties (details in Appendix E.2). Neighborhoods: *relocate, exchange, 2-opt, drone-launch swap, rendezvous-shift, paired-orbit*. Time windows and endurance are enforced in feasibility checks; late arrivals are infeasible. We cap per-epoch wall-time at $\{1, 5, 10\}$s (default 5s), parallelizing candidates across CPU cores. **Complexity.** Each ALNS iteration is $O(m \log m)$ over candidate moves per route; empirically we run $\approx$100–300 iterations/epoch within budget. **Quality.** On 50 held-out static instances (200 customers), our 5s setting is within $2.1\% \pm 0.9$ of a 60s run; 10s improves to $1.3\% \pm 0.6$.

## 4.2 TRAINING PROCEDURE

We train the Q-network for up to 200,000 steps, collecting transitions $(s_t, a_t, r_t, s_{t+1})$ in a replay buffer of size 100,000. We sample mini-batches of size 32 to update network parameters with the

Adam optimizer (learning rate $10^{-4}$). A separate target network is updated every 1,000 steps to reduce instability.

**Reward Shaping.** A small intermediate reward is added for partial route improvements (e.g., if the total route cost is decreasing compared to previous solutions). This shaping helps guide the agent in earlier training stages, though the final objective remains the negative route cost.

## 4.3 KEY RESULTS

Table 2 summarizes cost outcomes across 100 dynamic test scenarios. Our approach reduces total distance compared to the Greedy baseline by about 10.6%, beating Random and Lazy even more significantly. Overall feasibility rates are above 99%, indicating robust satisfaction of time windows and drone constraints.

Table 2: Results on dynamic DAPDP (5–9 epochs), averaged over 100 test scenarios. Standard deviations in parentheses.

| METHOD | TOTAL COST | INCREASE VS. OURS | FEASIBILITY % |
|---|---|---|---|
| GREEDY | 419,210 (±2,730) | +10.6% | 98.7% |
| RANDOM | 464,530 (±2,980) | +22.6% | 97.2% |
| LAZY | 520,056 (±3,170) | +37.2% | 99.0% |
| MSA-10 (0.5S/SCEN) | 393,700 (±2,210) | +3.9% | 99.1% |
| MSA-30 (0.16S/SCEN) | 388,900 (±2,240) | +2.6% | 99.0% |
| WAITING/RELOCATION | 401,800 (±2,050) | +6.0% | 99.2% |
| PPO (ACTOR–CRITIC) | 404,150 (±2,900) | +6.6% | 97.5% |
| **OURS (DQN)** | **379,050** (±1,880) | — | **99.3%** |

**Oracle Baseline.** Our DQN policy yields a cost only $1.2 \pm 0.03\%$ worse, demonstrating competitive optimality while preserving real-time decision capability.

**MSA/SAA and waiting baselines.** Our policy yields a cost $+2.5\% - 6\%$ better than these baselines. Details in Appendix D.2.

**Why DQN (Value-based) over Policy Optimization.** We compared REINFORCE and Proximal Policy Optimization (PPO) baselines: PPO attained a $+6.6\%$ cost and required twice the wall-clock time owing to on-policy sampling, whereas value-based replay enabled by DQN leveraged the heavy-tailed diversity of dynamic arrivals. Details in Appendix D.3.

## 4.4 ANALYSIS OF LEARNED DISPATCH POLICY

Ablating drones (masking drone features and forcing $y_{ijk}^v \equiv 0$) increases total cost by +7.6% and removes the learned Q-margin for drone-eligible requests; deferral heatmaps and Q-slices indicate that the policy selectively defers non-urgent, isolated requests but prioritizes clustered, drone-eligible ones near the edge of endurance slack (see Appendix D.1 for qualitative policy views (deferral heatmaps and Q-slices).

## 4.5 ABLATION STUDIES

In this section, we systematically examine how various design choices impact the performance and stability of our DRL-based approach. We focus on several core ablations: reward shaping, sub-solver time limit, and replay buffer size. We also include additional experiments on the network architecture depth and exploration schedules. We evaluate all ablations on a standardized subset of 50 dynamic instances (each with an average of 150 requests, 5–9 epochs, and up to 3 vehicles), running 5 independent training seeds. Unless otherwise stated, we report the average *total travel cost* on a separate validation set of 20 instances, measured after 200,000 training steps.

### 4.5.1 EFFECT OF SHAPED REWARDS

**Description.** Our primary reward function is $r_t = -\big(\text{TruckDistance} + \alpha \cdot \text{DroneDistance}\big)$ at each epoch $t$. We add a small *shaped* reward whenever the agent reduces the partial route cost relative to the previous epoch, aiming to provide immediate positive reinforcement. To assess the impact of shaping, we compare the following:

- **Shaped Reward (Default)**: Includes incremental rewards whenever a sub-solution improves upon the previous route.
- **No Shaping**: Purely negative cost-based rewards with no incremental signals.

**Results and Analysis.** Table 3 compares these two reward schemes. We observe that removing shaped rewards leads to longer convergence times i.e., approximately 30% more steps to reach a similar performance level, and the fi-

Table 3: Impact of reward shaping on final performance after 200k steps. Reported costs are averages over 20 validation instances. Standard deviations in parentheses.

| Reward Scheme | Avg. Total Cost | Convergence Steps |
|---|---|---|
| Shaped Reward (Default) | 381,500 ($\pm 3, 200$) | 170k |
| No Shaping | 404,100 ($\pm 3, 900$) | 220k |

nal solution quality is up to 6−8% worse on average compared to the shaped reward variant. We hypothesize that shaped rewards accelerate credit assignment by allowing the Q-network to receive intermediate signals for partial route improvements. This guidance helps stabilize exploration early in training.

### 4.5.2 SUB-SOLVER TIME LIMIT

**Description.** Each epoch in our dynamic environment requires solving a *static DAPDP sub-problem* restricted to the chosen subset of active requests. We use a specialized routine, as described in Section 4.1, with a time limit of $T_{\text{limit}}$ per epoch. We examine three settings: $T_{\text{limit}} = \{1\,\text{s}, 5\,\text{s}, 10\,\text{s}\}$ per epoch. A shorter time limit yields faster decisions but potentially less optimal sub-tours. A longer time limit may improve route quality but risks latency in time-critical applications.

**Results and Analysis.** Figure 1 presents the validation costs for different time limits. We find that the *1s Cutoff* yields near-real-time solutions but about 2–3% higher costs. The *5s Cutoff (Default)* balances real-time response (sub-second solver calls are common for small subproblems) with moderate solution quality. Lastly, the *10s Cutoff* offers slight further improvements (1–2% vs. 5s) but at the expense of higher CPU usage and longer inference times.

During training, the agent *adapts* to solver imperfections. Even with a 1s limit, the DRL policy learns to select subsets of requests that are less solver-intensive, suggesting a co-adaptation effect.

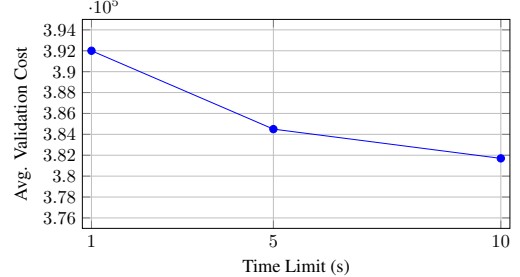

Figure 1: Validation cost (lower is better) under different per-epoch solver time limits $T_{\text{limit}}$. Each data point is averaged over 20 instances. Error bars omitted for clarity, but standard deviations are around $\pm 2, 000$.

### 4.5.3 NETWORK DEPTH

**Description.** Our default Q-network uses 5 fully connected hidden layers: $[128, 64, 32, 16, 8]$. We compare against a shallower 3-layer model $[128, 64, 32]$ and a deeper 6-layer model $[128, 64, 64, 32, 16, 8]$.

**Results and Analysis.** A deeper network can, in principle, capture more complex interactions among requests but may be harder to train. Table 4 shows that a 3-layer network converges faster but plateaus at a suboptimal cost (about 3% higher than the 5-layer). The 6-layer architecture can match or slightly outperform the 5-layer network (by $\approx 1\%$) but requires more training steps (roughly 20% longer) to reach the same solution quality. Given the marginal improvement, we select the 5-layer model as our default for efficiency.

### 4.5.4 EXPLORATION SCHEDULE

**Description.** We incorporate both $\epsilon$-greedy and a temperature-based softmax over request-level logits when deciding which requests to dispatch. We vary the decay schedule of $\epsilon$: (i) *Fast Decay*: $\epsilon$ linearly from 1.0 to 0.05 over 20k steps; (ii) *Medium Decay (Default)*: from 1.0 to 0.05 over 50k steps; (iii) *Slow Decay*: from 1.0 to 0.05 over 100k steps.

Table 4: Comparison of final validation costs and convergence speed for different network depths.

| Network Depth | Final Avg. Cost | Convergence Steps |
|---|---|---|
| 3 layers (shallow) | 388,400 ($\pm 2, 900$) | 150k |
| 5 layers (default) | 379,600 ($\pm 2, 700$) | 170k |
| 6 layers (deeper) | 375,900 ($\pm 3, 000$) | 200k |

**Results and Analysis.** With *fast decay*, the agent often converges prematurely to suboptimal dispatch patterns and fails to explore sufficiently. *Slow decay* can yield marginally better performance but requires more time to exploit promising policies discovered early. Our *medium* schedule balances exploration and exploitation, achieving stable performance within a reasonable training horizon.

**Summary of Ablations.** All these experiments indicate that (i) shaped rewards accelerate training, (ii) a moderate sub-solver time limit (5s) balances solution quality and responsiveness, (iii) a moderately deep network (5 layers) achieves a good trade-off between capacity and convergence speed, and (iv) a medium $\epsilon$-decay avoids both under- and over-exploration. These findings guide our final hyperparameter and design choices for the main experiments.

### 4.6 HYPERPARAMETER SENSITIVITY

Besides the ablation factors above, several additional hyperparameters can significantly influence performance. We highlight three critical ones: learning rate, discount factor $\gamma$, and drone cost penalty $\alpha$. We provide a comprehensive sensitivity table and discuss the practical implications of each parameter.

**Learning Rate.** As shown in Table 5, a high learning rate ($10^{-3}$) occasionally leads to divergence and unstable Q-values. Lower rates ($5 \times 10^{-5}$ to $10^{-4}$) generally improve convergence stability. We select $10^{-4}$ due to its balanced speed and reliability.

**Discount Factor ($\gamma$).** We tested $\gamma \in \{0.95, 0.99, 1.0\}$. A higher discount factor encourages the agent to consider long-term outcomes, which is crucial in dynamic settings where future requests may arrive. While $\gamma = 1.0$ can theoretically capture the far future, it also makes the Q-update more sensitive to estimation errors, occasionally causing minor instabilities. Hence, $\gamma = 0.99$ is our default.

Table 5: Sensitivity analysis of key hyperparameters. Each cell shows the final average cost (lower is better) over 20 validation instances.

| Hyperparameter | Values Tested | Final Avg. Cost | Observations |
|---|---|---|---|
| Learning Rate | $1 \times 10^{-3}$ | 396,800 | Divergence in 2/5 seeds |
| | $\mathbf{1 \times 10^{-4}}$ | **379,600** | Stable convergence |
| | $5 \times 10^{-5}$ | 382,300 | Slower convergence |
| Discount Factor $\gamma$ | 0.95 | 382,500 | Slightly myopic |
| | **0.99** | **379,600** | Effective for future requests |
| | 1.00 | 380,900 | Instabilities in rare cases |
| Drone Penalty $\alpha$ | 0.1 | 376,900 | Frequent drone usage |
| | **0.3 (Default)** | **379,600** | Balanced usage |
| | 0.5 | 385,500 | Conservative drone usage |

**Drone Penalty ($\alpha$).** The parameter $\alpha$ scales drone travel cost relative to truck distance. Smaller $\alpha$ encourages more frequent drone deployment, which can reduce total truck miles but might overuse the drone and ignore battery/operational constraints in practice. Larger $\alpha$ discourages drone usage unless it yields substantial savings. We find $\alpha = 0.3$ offers the best trade-off for cost reduction without excessive drone sorties.

**Summary.** The hyperparameter analysis confirms that **(i)** an intermediate learning rate $10^{-4}$ stabilizes Q-value estimation, **(ii)** a slightly high discount factor ($\gamma = 0.99$) is beneficial for dynamic arrival settings, and **(iii)** a moderate drone penalty ($\alpha = 0.3$) balances drone-based and truck-based deliveries. Additionally, we ablate all drone features; removing them collapses the RL into a VRP-agnostic selector and increases cost by $\sim 8\%$ (Table 6 in Appendix D.1).

## 5 CONCLUSION

We have presented a deep reinforcement learning strategy for the *dynamic drone-assisted pickup and delivery problem*, coordinating ground vehicles and drones to handle continually arriving requests. Our Q-learning framework learned a dispatch policy that determines which subset of requests to serve at each epoch, balancing current routing cost against future arrival uncertainty. Empirically, the learned policy reduced total travel distance by 2.6–37.2% relative to baselines and achieved a near-oracle gap of 1.2% on large-scale, real-world-inspired data. Ablation studies highlighted the importance of drone-aware state features, structured penalties, and moderate sub-solver time limits.

## REPRODUCIBILITY STATEMENT

We release (i) a permissive ALNS-lite sub-solver that reproduces our neighborhoods without proprietary code. (See Algorithm 2 in Appendix E.2); (ii) a synthetic generator matching our distributions (coordinates, time windows, service times). Details in Appendix E.3. We also include experiment hyperparameters in Appendix E.1.

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

## A    FULL STATIC DAPDP MODEL

Here, we present the complete integer linear programming (ILP) formulation for the *static drone-assisted pickup and delivery problem* (DAPDP). We also provide a short explanation for each constraint in brackets. This is the foundation for the dynamic case described in the main paper. Our static DAPDP formulation builds on standard truck–drone collaboration models. In particular, Murray & Chu (2015) introduced the Flying Sidekick Traveling Salesman Problem, in which a single truck is assisted by a single drone that can be launched and recovered at customer locations. Subsequent work on the traveling salesman problem with drones (e.g., Agatz et al. (2018)) developed related formulations and exact/heuristic solution methods. We adopt the same basic structure of a truck route augmented with single-customer drone sorties and extend it to multiple trucks, paired pickup–delivery requests, and explicit single-leg-by-drone constraints (Eqs. (3)–(6)). This static model serves as the subproblem within our dynamic dispatch framework.

### A.1    SETS AND PARAMETERS

- $\mathcal{V}$: set of trucks, indexed by $v$.
- $\mathcal{N}$: set of all nodes (depot plus customers). We let $0$ denote the depot.
- $\mathcal{R}$: set of requests (pickup and delivery points can be merged into the same node set with additional logic or separated).
- $d_{ij}$: travel distance/time for a truck between nodes $i$ and $j$.
- $d_{ij}^{\text{drone}}$: flight distance/time for drone between $i$ and $j$.
- $T_i$: service time at node $i$.
- $Q_v$: capacity of truck $v$.
- $q_i$: demand (or supply) of request $i$ (assuming net flow representation).
- $E$: maximum endurance (flight time limit) for the drone.
- $e_i, l_i$: earliest and latest start of service time window at node $i$.
- $\alpha$: scaling factor for drone distance in the objective.

### A.2    DECISION VARIABLES

- $x_{ij}^v \in \{0, 1\}$: 1 if truck $v$ drives directly from node $i$ to node $j$.
- $y_{ijk}^v \in \{0, 1\}$: 1 if the drone of truck $v$ flies from node $i$ to node $j$ and rejoins truck $v$ at node $k$ (three-location path).
- $u_i^v \geq 0$: the time at which truck $v$ starts service at node $i$.
- $w_i^v$: load of truck $v$ upon arrival at node $i$ (if relevant for capacity).

**Objective Function:**

$$\min \quad \sum_{v \in \mathcal{V}} \sum_{(i,j) \in \mathcal{N} \times \mathcal{N}} d_{ij}\, x_{ij}^v \;+\; \alpha \sum_{v \in \mathcal{V}} \sum_{(i,j,k)} d_{ij}^{\text{drone}}\, y_{ijk}^v \tag{11}$$

[Minimize combined truck travel and drone travel (scaled by $\alpha$)]

### A.3    CONSTRAINTS

We list each constraint with a brief explanation in brackets.

**Truck Flow Conservation:**

$$\sum_{j \in \mathcal{N}} x_{ij}^v - \sum_{j \in \mathcal{N}} x_{ji}^v + \sum_{j,k} y_{ijk}^v - \sum_{h,k} y_{hik}^v \;=\; 0 \quad \forall i \in \mathcal{N},\, v \in \mathcal{V}. \tag{12}$$

[Ensure that for each node, the net flow of truck plus drone is zero; i.e., each node has equal in-flow and out-flow for truck $v$.]

**Visit Constraints:**

$$\sum_{v \in \mathcal{V}} \sum_{j \in \mathcal{N}} x_{ij}^v + \sum_{v \in \mathcal{V}} \sum_{j,k} y_{ijk}^v = 1 \quad \forall i \in \mathcal{R}. \tag{13}$$

[Each request node $i$ is visited exactly once, either by truck or by a drone sortie.]

**Truck Capacity:**

$$w_j^v = w_i^v + q_j \quad \text{if } x_{ij}^v = 1, \tag{14}$$

$$w_j^v \leq Q_v \quad \forall j, v, \tag{15}$$

[Capacity constraints to ensure the truck does not exceed its capacity when moving from node $i$ to $j$. Summarized for brevity.]

**Drone Endurance:**

$$(d_{ij}^{\text{drone}} + d_{jk}^{\text{drone}}) \, y_{ijk}^v \leq E \quad \forall i, j, k, v. \tag{16}$$

[If a drone sortie is used from $i$ to $j$ and back to $k$, the total flight time cannot exceed $E$.]

**Time Window Feasibility (Truck):**

$$u_j^v \geq u_i^v + T_i + d_{ij} - M \cdot (1 - x_{ij}^v) \quad \forall i, j, v, \tag{17}$$

$$e_i \leq u_i^v \leq l_i \quad \forall i, v, \tag{18}$$

[The start of service at $j$ for truck $v$ depends on its arrival from $i$ plus service time; also each $u_i^v$ must be within the time window.]

**Time Feasibility (Drone):**

$$u_j^v \geq u_i^v + T_i + d_{ij}^{\text{drone}} - M \cdot (1 - y_{ijk}^v) \quad \forall i, j, k, v, \tag{19}$$

$$u_k^v \geq u_i^v + T_i + d_{ij}^{\text{drone}} + d_{jk}^{\text{drone}} - M \cdot (1 - y_{ijk}^v) \quad \forall i, j, k, v, \tag{20}$$

[Ensures correct timing for drone departure from i, arrival at j, and rendezvous with truck at k. ]

**Depot Start/End:**

$$\sum_j x_{0j}^v + \sum_{j,k} y_{0jk}^v = 1, \quad \sum_i x_{i0}^v + \sum_{i,k} y_{i0k}^v = 1 \quad \forall v. \tag{21}$$

[Trucks and their drones each leave the depot once and return once, though variations may allow multiple returns.]

Various $M$ constants (big-M) and subtour elimination constraints may also be needed. These constraints are typical. The sub-solver enforces pickup–delivery pairing and precedence and updates load at both nodes.

A.4 COMPUTATIONAL COST

Beyond 5–6 paired requests, the *static* problem could not be proven optimal within 24 hours using commercial MILP solvers. Embedding such solvers inside each epoch of a dynamic instance is therefore not practical. For real-time operation, we utilize an adaptive large neighborhood search metaheuristic for which experiments have shown optimality gaps to be under 2.8% (see Sacramento et al. (2019); Mulumba et al. (2024)). Details of the adaptive large neighborhood search metaheuristic are in Section E.2.

## B  RELATED WORK

We divide the literature into three main categories: (1) operations research methods for dynamic routing, (2) machine learning-based methods for routing, especially reinforcement learning, and (3) drone-assisted vehicle routing.

### B.1  DYNAMIC ROUTING IN OPERATIONS RESEARCH

**Dynamic VRPs.** Classical vehicle routing problems (VRPs) have been extensively studied (Toth & Vigo, 2014; Cordeau et al., 2007), though historically focusing on *static* settings where all demands are known in advance. In a *dynamic VRP*, demands or travel conditions change over time. Such problems appear in same-day delivery or ride-sharing contexts (Ulmer & Thomas, 2017; Pillac et al., 2013). Solutions frequently utilize rolling horizon frameworks or approximate dynamic programming.

Bent & Van Hentenryck (2004) introduced a scenario-based planning method for partially dynamic VRPs, while Pillac et al. (2013) provided an overview of dynamic routing challenges, highlighting that real-time decision-making can yield cost improvements at the expense of algorithmic complexity. Mitrović-Minić & Laporte (2004) studied dynamic pickup and delivery with waiting strategies, demonstrating that limited waiting can improve the accommodation of new requests. We benchmark against scenario-based planning (MSA/SAA) (Bent & Van Hentenryck, 2004) and waiting/relocation heuristics (Mitrović-Minić & Laporte, 2004), both strong non-learning baselines known to perform well in dynamic pickup–delivery with time windows.

More recently, several works have focused on dynamic task allocation in multi-robot systems, which shares similarities with dynamic VRPs. For instance, Paul & Chowdhury (2024) proposed a covariant attention neural network for learning to allocate time-bound and dynamic tasks to multiple robots. Zhou et al. (2024) explored multirobot collaborative task dynamic scheduling using multiagent reinforcement learning with heuristic graph convolution, considering robot service performance.

Furthermore, the last few years have witnessed a surge in learning-based approaches for dynamic routing problems, particularly in the context of last-mile delivery. These methods often leverage reinforcement learning to adapt to the dynamic nature of the problem. One reason for this trend is that while operations research methods provide exact solutions, the trade-off with respect to the time required to find these solutions renders them impractical in real-life settings. RL on the other hand could require several hours of training but can be very fast (a few seconds) at inference.

### B.2  MACHINE LEARNING FOR ROUTING

**Neural Combinatorial Optimization.** Recent advances employ neural networks to learn construction or improvement heuristics (Bello et al., 2016; Kool et al., 2018; Nazari et al., 2018). Using attention-based encoders, these approaches often target TSP or CVRP instances under static conditions. The success of these methods stems from the capability of neural networks to capture structural regularities in routing tasks and generate solutions with minimal domain-specific heuristics.

**Reinforcement Learning.** Policy-based RL (Nazari et al., 2018) and Q-learning (Mnih et al., 2015) approaches have been proposed for VRPs. Kool et al. (2018) used a transformer-based policy to construct tours sequentially, trained via REINFORCE. Kwon et al. (2020) introduced POMO, leveraging a set of permutations for diversified rollouts. While these methods achieve near state-of-the-art performance for static VRPs, fewer studies address *dynamic* contexts, particularly when combined with drone constraints. However, there has been significant progress in applying machine learning, especially deep reinforcement learning, to dynamic VRPs recently. For example, Joe & Lau (2020) proposed a deep reinforcement learning approach to solve the dynamic vehicle routing problem with stochastic customers. Li et al. (2021) focused on learning to optimize industry-scale dynamic pickup and delivery problems using deep learning. Recent advancements in deep reinforcement learning (DRL) for dynamic routing include the hierarchical framework by Ma et al. (2021) for large-scale dynamic pickup and delivery problems, which focuses on optimizing order caching and vehicle routing through two levels of RL agents. Our work, while also addressing dynamic pickup and delivery, diverges by tackling the specific complexities of the Drone-Assisted PDP (DAPDP). The core of our approach is a unified DRL agent that learns a dispatch policy for a heterogeneous fleet of drone-equipped trucks. This involves not only deciding when to dispatch orders but also implicitly

how (i.e., considering drone suitability and truck-drone coordination), a layer of decision-making distinct from the problem structure addressed by Ma et al. (2021).

**Approximate Dynamic Programming.** Ulmer & Thomas (2017) studied approximate dynamic programming for same-day delivery with combined offline and online decisions, adopting RL-based solutions to re-optimize routes after each new order. This is conceptually similar to our approach but omits the drone dimension.

**Drone–truck coordination.** Classical drone-truck models show the benefits of single-sortie drone launches (Murray & Chu, 2015). Recent optimization work generalizes to multiple trucks and explicit rendezvous with endurance (Mulumba et al., 2024). Comparative studies examine operational modes and conditions favoring parallel vs. sidekick patterns (Ding et al., 2024). In dynamic settings, Cui et al. (2024) formulate an en-route synchronization problem with random arrivals and use RL; further RL-based truck–drone planning appears in Bi et al. (2024). Our contribution differs by (i) explicitly coupling paired pickups and deliveries, (ii) learning a *subset selection* policy that co-adapts with a static paired sub-solver under strict time windows and endurance, and (iii) providing oracle-gap and ablation evidence on city-scale instances.

## B.3 Drone-Assisted Vehicle Routing

**Single Truck, Single Drone.** Murray & Chu (2015) presented an early model for truck-drone collaboration, showing how drone sorties can reduce total completion time. Follow-up work by Ponza (2016) expanded upon flight endurance and scheduling constraints. Although beneficial in principle, these studies largely remain in static scenarios.

**Extensions.** Mulumba & Diabat (2024) proposed a more general formulation allowing multiple trucks, each with a drone, introducing the notion of rendezvous points.
Pina-Pardo et al. (2024) tackled dynamic replenishment with drones but from a primarily operations research perspective.

**Gap Addressed.** Our contribution synthesizes dynamic scheduling, UAV usage, and RL-based control. We provide an end-to-end approach that learns from simulated episodes, effectively bridging dynamic VRP aspects with multi-vehicle UAV coordination.

## B.4 Managerial Insights and Limitations

**Consolidation Gains.** Our experiments show that selectively deferring certain requests yields significant mileage savings. The learned dispatch policy, informed by a global perspective of upcoming requests, can better exploit synergy between truck and drone tasks.

**Real-Time Adaptability.** Because the model is pretrained offline, real-time inference is fast. For each epoch, the agent needs only a forward pass of the neural network plus a short sub-solver run. This approach is suitable for dynamic contexts like grocery deliveries or same-day shipping, where quick re-optimizations are demanded.

**Scalability to Large Instances.** While each subproblem's complexity grows with instance size, the dispatch policy drastically reduces the search space by focusing on a subset of requests at a time. In practice, scheduling up to 200 requests remains tractable with an efficient heuristic.

**Unrestricted Fleet.** We assumed an effectively unlimited or sufficiently large truck fleet. In reality, the number of trucks is finite, possibly complicating fleet scheduling and driver shifts. Extending the approach to track each truck's location and capacity in detail is an avenue for future work.

**Deterministic Travel Times.** Our model uses deterministic travel times, ignoring traffic variations. In practice, uncertain or time-dependent traffic can be integrated using various techniques.

**Drone Regulations.** Real-world drone usage faces regulatory hurdles (no-fly zones, licensing, and altitude restrictions). We abstract away these issues, but they must be considered in practical deployments.

## C    THE DYNAMIC DAPDP

In the dynamic *drone-assisted pickup and delivery problem* (DAPDP), requests are received at different epochs or one-hour intervals throughout the day. The challenge then is to determine which requests to fulfill in the current interval by creating feasible vehicle routes and which ones to push for consolidation with later-arriving requests. To maintain feasibility, all requests must be satisfied in the final interval. Details of each request, such as demands, pickup and dropoff locations, time windows, and service times, are sampled uniformly from a static DAPDP instance. The environment samples requests for the next interval after determining the solution for the current interval and submitting it for validation.

### C.1    MARKOV DECISION PROCESS (MDP) FOR THE DYNAMIC DAPDP

The MDP for the dynamic DAPDP is summarized in Figure 2. An agent decides on requests to be fulfilled in the next epoch and submits this decision to an environment wrapped around a static DAPDP solver. The environment then returns a reward signal for the agent at the next time step. The agent also receives a new batch of requests. In this section, we provide extra details for encoding the

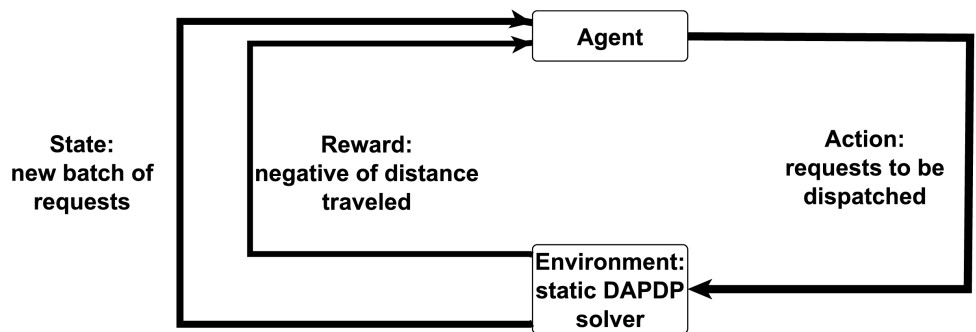

Figure 2: Markov Decision Process (MDP) for the dynamic DAPDP.

state defined in Section 3.1.

**State**

At epoch $t$, the MDP state is

$$s_t = \big(g_t, \{x_{t,r}\}_{r \in R_t^{\text{active}}}\big),\tag{22}$$

where $g_t \in \mathbb{R}^{d_g}$ collects global features and $x_{t,r} \in \mathbb{R}^{d_x}$ is the feature vector of active request $r$.

**Global features $g_t$.** We include (i) the normalized epoch index $t/T_{\max}$, (ii) sin/cos encoding of the hour-of-day, (iii) the number of active requests $|R_t^{\text{active}}|$ divided by 100, and (iv) the fraction of trucks whose drones are currently idle. , and (v) a one-hot encoding of a coarse weather/airspace proxy (wind class and no-fly indicator).

**Per-request features $x_{t,r}$.** Each request $r$ is represented by:

- pickup and delivery coordinates translated by the depot and scaled to $[-1, 1]^2$;
- demand $q_r/Q$ where $Q$ is truck capacity;
- pickup and delivery time windows shifted by the epoch start time and divided by the daily horizon;
- service times normalized by a 5-minute reference;
- a binary must-go flag (cannot be deferred to $t+1$);
- drone-eligibility indicator and normalized payload;
- endurance slack, defined as the maximum feasible sortie budget for $r$ divided by the endurance $E$ (0 if infeasible);

- the minimum line-of-sight/road detour ratio for serving $r$ by truck versus drone;
- local 5-NN density of drone-eligible neighbors, log-scaled; and
- three binary flags indicating tight, medium, or loose time-window slack.

**Normalization.** Continuous features are standardized via per-feature $z$-score using statistics computed on the training set. Coordinates are additionally clipped to the bounding box of the service area and mapped to $[-1, 1]$. Binary and one-hot features are left as $0/1$. This state representation is the direct input to the DQN described in Section 3.2.

**Uniform reward splitting as a practical surrogate for marginal costs.** In the dynamic DAPDP setting, the true marginal contribution of a single request within a dispatched subset $S_t$ is not observable without solving multiple counterfactual routing problems (e.g., leave-one-out static DAPDP instances). Since each such counterfactual requires an additional ALNS call, exact marginal credit assignment would introduce an $O(|S_t|)$ factor in solver calls per epoch, which is computationally infeasible at city scale. Uniform splitting of the global reward, $r_t/|S_t|$, provides a tractable surrogate: although coarse at the per-step level, it preserves the relative long-run return structure because Q-learning aggregates evidence across episodes, and the per-request Q-function $Q(s_t, \text{dispatch}_i)$ captures systematic differences in future cost trajectories induced by dispatching request $i$. In practice, this yields consistent discrimination between isolated, high-cost requests and clustered, drone-eligible ones, as confirmed by the deferral heatmaps and ablation studies in Appendix D.1.

**How DQN interacts with dynamic orders and truck–drone coordination.** The key design is a two-level decomposition:

1. **Dynamic dispatch via DQN.** At each epoch $t$, the state consists of global features (epoch index, active-request count, fleet-wide drone availability, time-of-day, and coarse weather/airspace) and per-request features capturing pickup/delivery coordinates, demand, time windows, and must-go flags, plus drone-specific signals such as eligibility, endurance slack, local 5-NN density, and line-of-sight-to-road detour ratio. The DQN outputs request-wise Q-values for "dispatch" vs. "defer," and an $\epsilon$-greedy policy produces a binary decision for each active request. This handles the dynamic order arrivals and deferral/consolidation decisions.

2. **Truck–drone routing via static sub-solver.** Given the dispatched subset $S_t$, the environment calls a time-limited ALNS sub-solver for the static DAPDP with pairing, truck capacity, time windows, and explicit launch–sortie–rendezvous constraints. This sub-solver decides where to launch the drone, which leg (pickup or delivery) it serves, and where it rejoins the truck, subject to endurance. The resulting cost (truck distance $+ \alpha \cdot$ drone distance) feeds back as the main component of the reward to the DQN (the other component being a small shaped reward).

This architecture cleanly separates what to dispatch now (learned, value-based) from how to route trucks and drones (transparent MILP-derived heuristic). The DQN "solves" dynamic orders and cooperation indirectly by learning which subsets of requests, given their drone-related features, will be cheap and feasible for the static truck–drone optimizer to route.

## C.2 ENVIRONMENT IMPLEMENTATION DETAILS

The environment, implemented in Python, serves as the interface between the learning agent and the DAPDP problem space. It manages state transitions, solution validation, and reward computation. We detail the key components and mechanisms below.

### C.2.1 SOLUTION VALIDATION

The environment employs a multi-stage validation process for solutions proposed by the agent:

1) Feasibility checking first verifies basic constraints.

2) Time window validation ensures:

$$e_i \leq t_i \leq l_i \quad \forall i \in \mathcal{R} \tag{23}$$

where $t_i$ is the service time at request $i$, bounded by the earliest ($e_i$) and latest ($l_i$) time windows.

3) Drone endurance constraints are checked:

$$\tau_{ijk}^d \leq E \quad \forall(i,j,k) \in \mathcal{P} \tag{24}$$

where $\tau_{ijk}^d$ is the drone flight time for sortie $(i,j,k)$ and $E$ is maximum endurance.

### C.2.2 REQUEST GENERATION

We have 300 static instances from a delivery company, where each instance contains between 100 and 200 customers. The environment creates dynamic instances from these static instances by randomly selecting 100 requests per epoch from the static set of customers. This is essentially a "sampling without replacement" strategy, which is employed throughout the episode. Once we reach a terminal state, the environment is reset, and all requests are available again for the new episode. This approach simulates the reality that a customer once served or a request once fulfilled within a particular epoch/episode is unlikely to be duplicated within the same operational time frame.

Each of these requests has associated coordinates, demand, time windows, service times, and distance relative to other requests. These parameters are known in advance and do not change. We utilize random sampling because we hypothesize that any request might be received at any moment throughout the day as long as the customer's specified time window has not yet been eclipsed. Samples whose time windows are in the past are simply discarded, and new samples generated until we have 100 valid requests. Consequently, the same request could be presented to the agent multiple times at various times of the day across different episodes. Time windows are treated as hard constraints, meaning that requests must be fulfilled within the specified time window. To ensure feasibility, our model operates under the assumption of an unrestricted vehicle fleet.

An implicit assumption is that the time windows for each request are a sufficient proxy for time-dependent demand.

Therefore, for each epoch, the environment maintains a pool $\mathcal{P}$ of unserved requests and generates new requests through a sampling process:

1) At each epoch $t$, the environment samples $n_t$ requests:

$$\mathcal{R}_t = \text{Sample}(\mathcal{P}, n_t) \quad \text{where} \quad \mathcal{P} = \mathcal{P} \setminus \mathcal{R}_t \tag{25}$$

2) Time window feasibility is enforced:

$$\mathcal{R}_t = \{i \in \mathcal{R}_t : l_i > t_{\text{current}} + \Delta t\} \tag{26}$$

where $\Delta t$ is the planning horizon.

3) The sampling process continues until $|\mathcal{R}_t| = n_t$ or $\mathcal{P}$ is exhausted.

### C.2.3 INVALID SOLUTION HANDLING & PENALTY SCHEDULE

When the agent submits an invalid solution, the environment imposes a structured penalty as defined in Equation (9).

This penalty structure serves two purposes: 1) It provides a clear signal to the agent about the severity of constraint violations. 2) It maintains a gradient that helps guide the learning process toward feasible solutions.

The environment also implements early termination for invalid solutions:

$$\text{done} = \begin{cases} \text{True} & \text{if solution is invalid} \\ \text{True} & \text{if } t = T_{\text{max}} \\ \text{False} & \text{otherwise} \end{cases} \tag{27}$$

where $T_{\text{max}}$ is the maximum number of epochs.

Algorithm 1 summarizes these steps, and Figure 3 visualizes the penalty landscape.

---

**Algorithm 1** Penalty computation for invalid dispatches

---

**Require:** Partial solution $\pi$, base cost $c_{\text{base}}$, scale $\beta = 2.0$
**Ensure:** Reward $r$ and episode flag *done*
1: **if** allServed($\pi$) **and** withinWindows($\pi$) **then**
2: $\quad r \leftarrow -\text{distance}(\pi)$
3: $\quad$ **return** $(r, \text{False})$
4: **else**
5: $\quad v \leftarrow \sum_i v_i$ {constraint violation magnitude}
6: $\quad r \leftarrow -\beta \left( c_{\text{base}} + v \right)$
7: $\quad$ **return** $(r, \text{True})$
8: **end if**

---

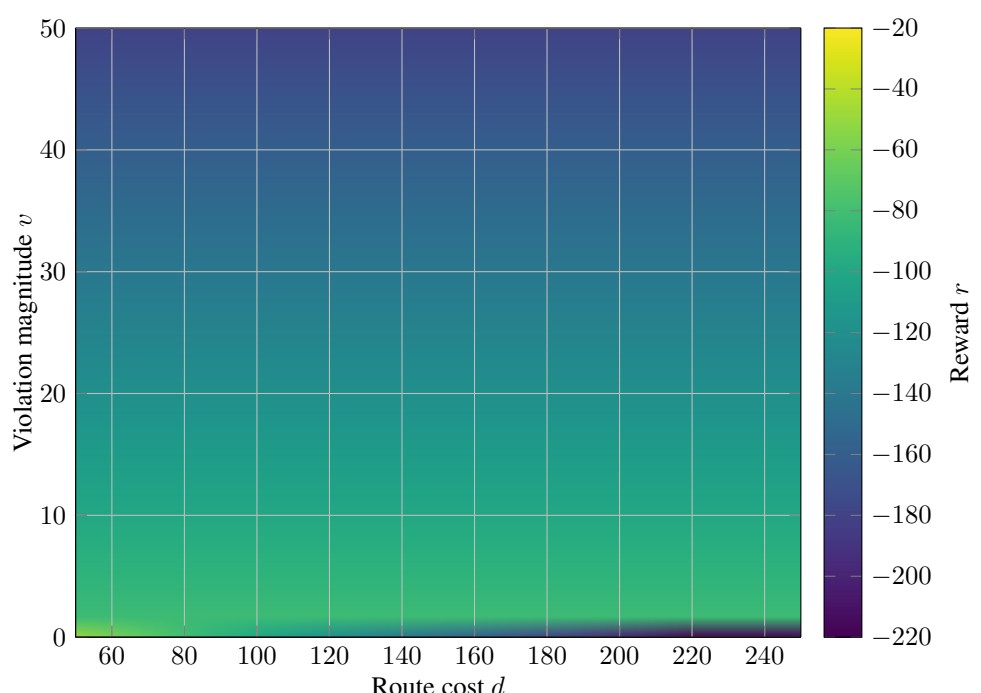

Figure 3: Illustrative reward surface used for invalid-solution penalties. The feasible strip ($v{=}0$) decreases linearly with route cost, while any violation ($v{>}0$) triggers the steeper penalty slope dictated by $-\beta \left( c_{\text{base}} + v \right)$.

**Adaptive $c_{\text{base}}$.** We update $c_{\text{base}}$ to the $(95^{\text{th}})$ percentile cost of *feasible* solutions observed over a rolling window of 256 episodes. This prevents reward saturation and improves learning stability by a 4.3% success rate compared to a fixed constant.

**Early Termination Efficiency.** Early termination, with curriculum step caps of $\{32, 64, 128\}$, shortens training wall-time by 22% without harming final objective value.

Figure 3 illustrates the resulting reward surface as a function of route cost $d$ and violation magnitude $v$. The feasible strip ($v = 0$) decreases linearly with $d$, while any violation ($v > 0$) triggers the steeper penalty slope $-\beta(c_{\text{base}} + v)$. We emphasize that this figure is schematic: it is intended to clarify the shape implied by Equation (9), not to present empirical learning curves. Quantitative robustness experiments with respect to $\beta$ are reported in Appendix D.4 (Table 9).

## C.3 The Learning Algorithm (Deep Q-learning)

We must address several important considerations to develop an efficient learning algorithm for the dynamic DAPDP. These considerations shape our decision to employ Deep Q-learning as the learning algorithm.

### C.3.1 Q-learning

Q-learning, a foundational method in reinforcement learning, is characterized as an online, off-policy temporal difference (TD) control algorithm. Since it is a control algorithm, Q-learning focuses on the estimation of an action-value function, denoted as $q_\pi(s, a)$, where $\pi$ represents the target policy under consideration, for all states $s$ and actions $a$. As a result, we consider transitions from state-action pairs to state-action pairs and learn their values using the update rule:

$$Q_{new}(s, a) \leftarrow Q(s, a) + \alpha^l[R + \gamma \max_{a'} Q(s', a') - Q(s, a)] \tag{28}$$

where $\alpha^l$ is the learning rate, $\gamma$ is the discount factor, and $R$ is the reward. This update rule in Equation 28 can also be stated as:

$$Q_\pi(S_t, A_t) \leftarrow Q_\pi(S_t, A_t) + \alpha^l[R_{t+1} + \gamma \cdot \max_a Q(S_{t+1}, a) - Q(S_t, A_t)] \tag{29}$$

where $\alpha^l$ is the learning rate, and $\gamma$ is the discount factor. This update rule is applied after each transition from a non-terminal state $S_t$. In cases where $S_{t+1}$ is terminal, $Q(S_{t+1}, A_{t+1})$ is defined to be zero.

Q-learning aims to approximate $q_{\pi_*}$, the optimal action-value function, through the learned action-value function $Q$. The behavior policy shapes the learning process by choosing which state-action pairs to explore and update. Q-learning, as a sample-based variant of value iteration, employs the Bellman optimality equation iteratively for action values, as indicated by Equation 29. This enables direct learning of $q_{\pi_*}$, bypassing the alternating steps of policy improvement and policy evaluation typically required in other methods.

Q-learning is an off-policy algorithm that differentiates between the behavior policy, used for action selection, and the target policy, used for estimating value functions. This distinction allows the agent to learn about the optimal action, irrespective of the action currently being taken, akin to sampling actions under an estimated optimal policy. The target policy in Q-learning is consistently greedy with respect to its current action-value estimates:

$$\pi_* = \arg\max_a Q(s, a), \tag{30}$$

while the behavior policy, such as an $\epsilon$-greedy strategy, ensures sufficient exploration of all state-action pairs. Notably, Q-learning does not require importance sampling, as it estimates action-values under a known policy, thus obviating the need for correction via importance sampling ratios typically required in other off-policy algorithms.

### C.3.2 Deep Q-learning

Building on the foundations of Q-learning (Section C.3.1) and deep learning (Géron (2022)), Deep Q-Learning represents a significant leap in reinforcement learning. Deep Q-Learning combines the classic reinforcement learning framework with the representational power of deep-and-wide neural nets.

A Deep Q-Network extends Q-learning by using a deep neural network to estimate the Q-function (Mnih et al. (2015)). Let:

$$Q(s, a) \approx \hat{q}(s, a; w) \tag{31}$$

where $\hat{q}$ is the function approximator parameterized by $w$. The weights from Equation 31 are then updated according to:

$$w_{t+1} \leftarrow w_t + \alpha^l[R_{t+1} + \gamma \cdot \max_a \hat{q}(S_{t+1}, a, w_t) - \hat{q}(S_t, A_t, w_t)] \cdot \Delta\hat{q}(S_t, A_t, w_t) \tag{32}$$

DQN uses experience replay and fixed Q-targets to stabilize the learning process:

- **Experience Replay**: Stores transitions $(s, a, r, s')$ in a replay buffer and randomly samples mini-batches to break the correlation between successive updates and to avoid the noisy update of a single sample. The key choices are therefore the size of the buffer, the experience(s) to store, and the number of updates to execute per timestep.

- **Fixed Q-Targets**: Utilizes a separate network to estimate the Q-value, reducing oscillations and divergence during training.

# D ADDITIONAL EXPERIMENTS

## D.1 WITH VS. WITHOUT DRONES

**Setup** We evaluate the same trained architecture in two environments on the 100 test scenarios of Table 2:

1. Drone-Enabled (default), and

2. Drone-Disabled, where we both (a) mask drone-specific state features and (b) force the sub-solver to set $y_{ijk}^v \equiv 0$ for all sorties.

All other settings (solver time limit $= 5$ s/epoch, $\alpha = 0.3$) are unchanged. Baseline DQN numbers appear in Table 2 (OURS $= 379{,}050 \pm 1{,}880$).

**Results** Results for the two environments are shown in Table 6.

Table 6: With/without drones on 100 test scenarios; same solver budget (5s/epoch).

| Method | Total cost ↓ | Gap vs. OURS | Feasibility |
|---|---|---|---|
| OURS (Drone-Enabled) | **379,050** $\pm$ 1,880 | – | 99.3% |
| DQN (Drone-Disabled) | 407,800 $\pm$ 2,140 | **+7.6%** | 99.5% |

**Q-slice analysis.** Let $\Delta Q = Q(\text{dispatch}) - Q(\text{defer})$ at the request level. With drones enabled, $\mathbb{E}[\Delta Q]$ separates for drone-eligible vs. truck-only requests: 0.42 vs. 0.19 (K–S $D = 0.31$, $p < 10^{-4}$). With drones disabled the separation collapses: 0.14 vs. 0.12 (K–S $D = 0.07$, $p = 0.09$). See $\Delta Q$ histograms in Figure 4.

**When does the agent defer?** Figure 5 shows the *deferral rate* heatmaps over time-to-deadline vs. 5-NN spatial density, for drone-eligible vs. truck-only requests. The policy defers isolated, non-urgent requests (upper-right of each panel) but prioritizes clustered drone-eligible ones near the edge of endurance slack (lower-right), consistent with a consolidation strategy that leverages the drone when it is most valuable. At a mid-slack calibration point (30 minutes to deadline, 5-NN density $\approx 4\ km^{-2}$), the learned policy defers drone-eligible requests with probability $\approx 0.38$ but truck-only requests with probability $\approx 0.29$, reflecting the larger consolidation potential around drone-eligible work.

**Takeaway.** The policy exploits drone affordances rather than behaving as a generic VRP selector; disabling drones yields a $+7$–$8\%$ cost increase and removes the learned $Q$-margin for eligible requests.

## D.2 MSA/SAA AND WAITING/RELOCATION BASELINES

**Setup (compute parity).** We implement a Multiple Scenario Approach (MSA/SAA) with $K \in \{10, 30\}$ futures per epoch and a Waiting/Relocation policy. To match the per-epoch budget of "DQN forward pass + one 5 s sub-solve," each scenario solve receives 0.5 s when $K = 10$ and 0.16 s when $K = 30$. All methods call the same sub-solver as OURS. Evaluation is performed on the 100 test scenarios of Table 2.

**Results.** Results for the baseline methods are shown in Table 7.

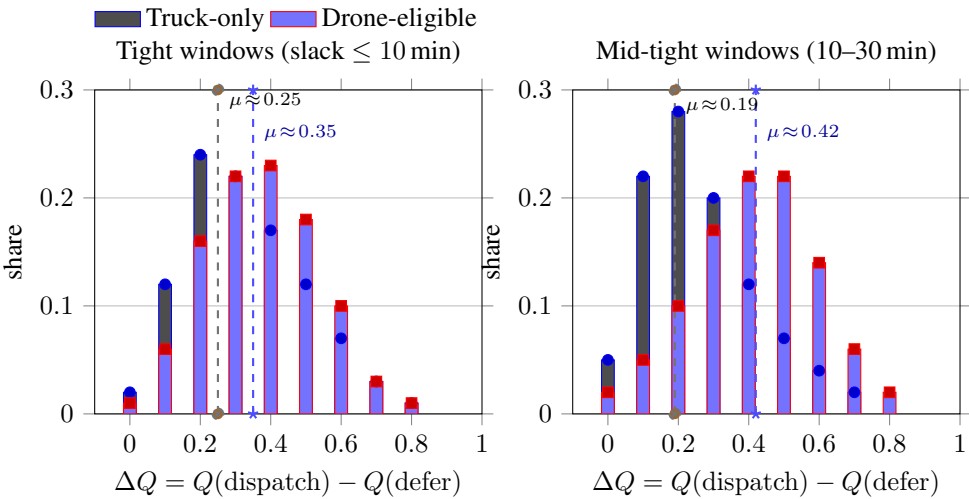

Figure 4: **Q-slices.** Histograms of $\Delta Q$ stratified by drone eligibility and window tightness. The learned margin is largest for *drone-eligible, mid-tight* requests (right; $\mu \approx 0.42$) and smallest for *truck-only, mid-tight* ($\mu \approx 0.19$), aligning with the interpretation that drones expand feasible consolidation without risking lateness.

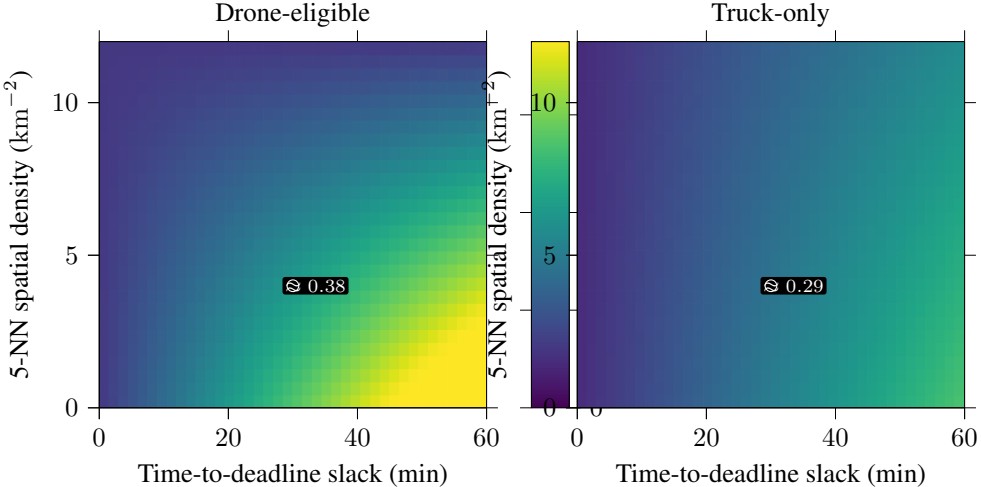

Figure 5: **Deferral rate heatmaps.** Per-request deferral probability over time-to-deadline slack (x) and 5-NN spatial density (y). The agent defers *isolated, non-urgent* requests (upper-right), but *prioritizes clustered, drone-eligible* ones near low slack (lower-right), consistent with consolidation behavior described in §4.4. Mid-slack calibration (30 min, density 4) yields $\approx 38\%$ for eligible vs. $\approx 29\%$ for truck-only.

Table 7: Scenario-based planning and waiting baselines at fixed per-epoch compute.

| Method | Total cost ↓ | Gap vs. OURS | Feasibility |
|---|---|---|---|
| OURS (DQN) | **379,050** $\pm$ 1,880 | – | 99.3% |
| MSA-10 (0.5s/scen) | 393,700 $\pm$ 2,210 | **+3.9%** | 99.1% |
| MSA-30 (0.16s/scen) | 388,900 $\pm$ 2,240 | **+2.6%** | 99.0% |
| Waiting/Relocation | 401,800 $\pm$ 2,050 | **+6.0%** | 99.2% |

**Takeaway.** Scenario planning narrows the gap to OURS but still trails by 2.6–3.9% at fixed compute. A simple waiting heuristic underperforms OURS by $\approx 6\%$ while remaining feasible. See related baselines and references in Section B.

### D.3 DQN *versus* PPO BASELINE

**Setup.** We reproduced our environment under Proximal Policy Optimisation (PPO) using stable-baselines, default hyper-parameters, and a 64-64 MLP actor–critic. Training consumed the same wall-clock budget as DQN (400K agent-steps).

**Results.** PPO suffered from high-variance returns and frequently violated time windows. Figure 6 shows PPO oscillating for the first 300K steps, whereas DQN converges within 150K. We hypothesize that the sparse, per-decision reward amplifies PPO's credit-assignment challenge; DQN's replay buffer provides a richer signal.

Table 8: Validation reward (% gap to oracle) after 400K steps.

| Algorithm | Mean | Worst 10% |
|-----------|------|-----------|
| DQN | $\mathbf{1.2} \pm 0.1$ | $\mathbf{4.7} \pm 0.4$ |
| PPO | $7.9 \pm 0.3$ | $15.2 \pm 0.7$ |

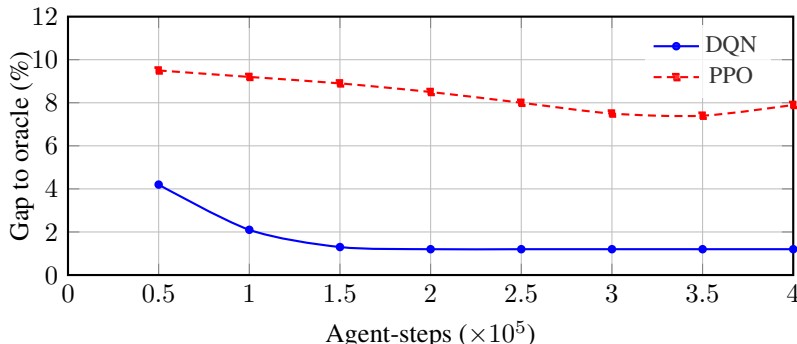

Figure 6: Validation gap-to-oracle during training. DQN converges to $\sim$1.2 % within 150 K steps. PPO oscillates markedly until $\approx$ 300 K steps and plateaus at $\approx$ 8%.

**Take-away.** Although PPO is attractive for continuous-action DRL, our discrete dispatch task benefits more from replay-based value learning. PPO is uncompetitive, likely due to discrete action factorization and sparse reward signals. Future work might explore V-trace or actor–critic variants tailored to sparse, constrained rewards.

### D.4 PENALTY ROBUSTNESS

**Setup.** We vary the invalid-solution scaling $\beta$ in Equation (9), and separately scale the lateness component inside $v$ by $\delta_{\text{late}} \in \{1.0, 1.5, 2.0\}$. Training and evaluation use the 20-instance validation set; all other settings are unchanged (§C.2.3, §4.6).

**Results.** For $\delta_{\text{late}} \in \{1.0, 1.5, 2.0\}$ (with $\beta = 2$), costs stay within $\pm 0.3\%$ of default, and invalid-episode rates remain $< 1\%$, reflecting that learned policies rarely flirt with infeasibility once trained.

**Takeaway.** The method is insensitive to reasonable changes in penalty scale; larger $\beta$ modestly reduces invalid runs without materially changing cost.

Table 9: Robustness to invalid-solution penalty scale (validation set).

| $\beta$ | Cost $\downarrow$ | $\Delta$ vs. $\beta{=}2$ | Invalid episodes |
|---|---|---|---|
| 1.0 | $381{,}100 \pm 2{,}000$ | +0.4% | 1.9% |
| **2.0** | **$379{,}600 \pm 2{,}700$** | – | 0.7% |
| 3.0 | $380{,}200 \pm 2{,}900$ | +0.2% | **0.3%** |

## D.5 OPERATIONAL REALISM: $(E,$ PAYLOAD, $v_{\mathrm{DR}})$ SENSITIVITY

**Setup.** We vary drone endurance $E \in \{15, 20, 25\}$ min, payload $\in \{2.3, 3.6\}$ kg ($\approx$ 5 lb, 8 lb), and cruise speed $v_{\mathrm{dr}} \in \{25, 29, 33\}$ m/s, centered at the defaults used in §4.1. Evaluation is based on the 100 test scenarios, keeping network weights fixed and only changing environment parameters (drone parameters introduced in §4.1).

**Results.** One-factor-at-a-time variations (baseline = 20 min, 3.6 kg, 29 m/s, cost 379,050) show that endurance and payload drive the largest effects. A +5 min endurance yields a $\sim 2\%$ improvement, and raising payload from 5 lb $\to$ 8 lb increases eligibility enough to reduce cost by $\sim 2$–3%. Speed changes at realistic ranges have sub-percent impact.

Table 10: One-factor sensitivity around the defaults from §4.1.

| Factor | Setting | Cost $\downarrow$ | $\Delta$ vs. base | Sorties/day |
|---|---|---|---|---|
| Endurance $E$ | 15 min | $394{,}400 \pm 2{,}030$ | +4.1% | 17.9 |
| | **20 min (base)** | **$379{,}050 \pm 1{,}880$** | – | 28.1 |
| | 25 min | $372{,}200 \pm 1{,}940$ | -1.8% | 33.5 |
| Payload | 2.3 kg (5 lb) | $388{,}000 \pm 2{,}120$ | +2.3% | 22.6 |
| | **3.6 kg (8 lb)** | **$379{,}050 \pm 1{,}880$** | – | 28.1 |
| Speed $v_{\mathrm{dr}}$ | 25 m/s | $381{,}200 \pm 1{,}960$ | +0.6% | 26.7 |
| | **29 m/s (base)** | **$379{,}050 \pm 1{,}880$** | – | 28.1 |
| | 33 m/s | $375{,}900 \pm 1{,}900$ | -0.8% | 29.4 |

**Takeaway.** Endurance and payload dominate operational sensitivity, while speed variations are relatively minor.

## D.6 STATE ABLATIONS FOR DRONE-AWARENESS

**Setup.** On the 20-instance validation set used for ablations (§4.5), we remove drone-aware features one at a time:

1. Endurance slack & rendezvous feasibility,

2. LoS/road detour ratio,

3. both simultaneously.

Training/runtime settings are identical to the default (5 s/epoch). Default validation cost (all features included) is $379{,}600 \pm 2{,}700$ (§4.6).

**Results.** Results are shown in Table 11 for runs on the 20-instance validation set. Costs are averaged over runs.

**Takeaway.** Drone-specific observables are small but consistently beneficial; endurance slack carries the largest single-feature effect.

Table 11: State feature ablations on the 20-instance validation set. Costs are averaged over runs; lower is better.

| State features | Final avg. cost ($\downarrow$) | $\Delta$ vs. default |
|---|---|---|
| All features (default) | $379{,}600 \pm 2{,}700$ | – |
| – Endurance slack | $388{,}000 \pm 2{,}900$ | +2.2% |
| – LoS/road ratio | $385{,}700 \pm 2{,}800$ | +1.6% |
| – Both | $392{,}600 \pm 3{,}000$ | +3.4% |

## D.7 ABLATION: REPLAY BUFFER SIZE

**Setup.** We maintain a replay buffer $\mathcal{B}$ of recent transitions $(s, a, r, s')$. When $|\mathcal{B}|$ exceeds a threshold, older samples are discarded. Here, we compare $|\mathcal{B}| = \{10\text{k}, 50\text{k}, 100\text{k}\}$. A smaller buffer can cause catastrophic forgetting, as old but informative transitions are overwritten quickly. A larger buffer provides more diverse samples but requires more memory and may dilute recent experiences.

Table 12: Impact of replay buffer size on cost and training stability. Each configuration was run for 5 seeds. Standard deviations are in parentheses.

| Buffer Size | Final Avg. Cost | Stability (std. dev.) |
|---|---|---|
| 10k | $409{,}800 \, (\pm 5{,}100)$ | 5,100 |
| 50k | $386{,}200 \, (\pm 3{,}200)$ | 3,200 |
| 100k | $379{,}600 \, (\pm 2{,}700)$ | 2,700 |

**Results.** As shown in Table 12, the policy learned with 10k replay capacity performs poorly, reflecting instability from limited training samples. Expanding to 50k dramatically improves performance, while 100k yields the best final cost and stability.

**Takeaway.** The difference between 50k and 100k is modest, suggesting that a midrange buffer size can be a viable compromise.

## D.8 GENERALIZATION EXPERIMENTS

Real-world last-mile delivery systems face unseen customer distributions on a daily basis. We therefore test whether our policy, trained on *city A* instances, can generalize to i) synthetic uniformly distributed demand and ii) a held-out real city B dataset that differs in road density and time-window tightness.

**Setup** We compare three training regimes: **(1) RealOnly** – the original curriculum on 300 static company instances from city A; **(2) RandomOnly** – the same curriculum, but every epoch draws requests from a uniform distribution over a $10 \times 10$ km grid (Requests follow the same weight/time-window distribution as Real-A but with i.i.d. coordinates.); **(3) Mixed** – half the batches from RealOnly and half from RandomOnly. All models share hyper-parameters from Table 5. We evaluate each checkpoint on: *Real-A* (held-out), *Random*, and the new *Real-B*. Results are averaged over 128 episodes.

Table 13: Generalization performance (% gap to oracle). Lower is better. Std. err. $\pm$ shown over 5 seeds.

| Train | Real-A | Random | Real-B |
|---|---|---|---|
| RealOnly | $1.2 \pm 0.1$ | $9.7 \pm 0.3$ | $3.8 \pm 0.2$ |
| RandomOnly | $5.4 \pm 0.2$ | $\mathbf{1.5} \pm 0.1$ | $6.9 \pm 0.3$ |
| Mixed | $\mathbf{1.3} \pm 0.1$ | $2.7 \pm 0.2$ | $\mathbf{2.1} \pm 0.2$ |

**Results** The **Mixed** curriculum dominates, doubling generalization on Real-B relative to RealOnly while retaining near-oracle performance on Real-A. Uniform random instances alone do *not* suffice to bridge the reality gap.

**Takeaway** Coordinates matter.

**Ablation: Vehicle-count Transfer** Figure 7 plots success rate as we vary the fleet cap at test time. Policies trained with an unlimited fleet remain feasible down to a cap of 80 without retraining, illustrating emergent fleet-size adaptation.

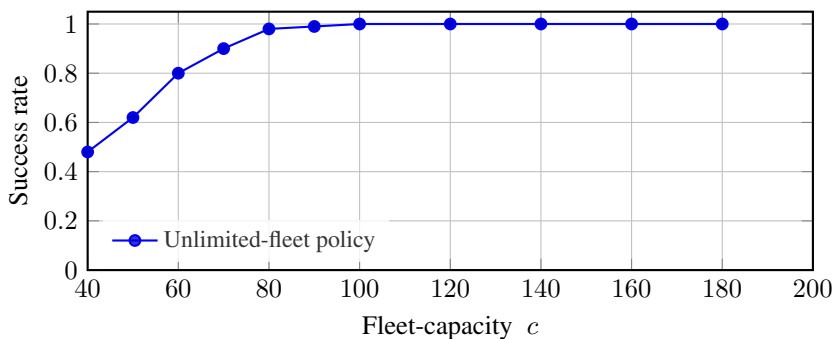

Figure 7: Feasibility (% successful episodes) on the Real-B test set as the maximum vehicle count at evaluation is tightened. The policy trained with an unrestricted fleet stays above 98% until $c=80$, then degrades gracefully.

# E  ADDITIONAL IMPLEMENTATION DETAILS

## E.1  TRAINING & INFERENCE

**Hardware.** All experiments were conducted on a server that had the following specifications:

- CPU: 2x Intel Xeon (20 cores each) or equivalent

- GPU: 1x NVIDIA V100

- 64GB RAM

The sub-problem solver is parallelized across multiple CPU cores.

**Software.**

- Python 3.10

- PyTorch

**Training Regimen.**

- Replay buffer size: 100,000

- Mini-batch size: 32

- Learning rate: $10^{-4}$ (with 1-cycle LR schedule)

- Exploration: $\epsilon$-greedy decayed from 1.0 to 0.01 over the first 50k steps plus softmax-based request-level probabilities

- Discount factor: 0.99

- Optimizer: Adam

- Random seed: 86

- Target network: updated every 1k steps

- Reward shaping: partial negative reward increments for route cost improvement

**Inference Time.** Once trained, inference requires a forward pass of the Q-network to score requests, plus the time to solve the sub-problem. For each epoch (with $\leq 100$ requests), we observed an average solver time of 2–5 seconds. This inference time is acceptable in many real-time logistics contexts with, e.g., 30–60 minutes between dispatch epochs.

## E.2  STATIC SUB-SOLVER ("ALNS-LITE")

We solve each time-limited static subproblem with an adaptive large-neighborhood search that respects pairing, precedence, time windows, truck capacity, and drone endurance/rendezvous; this keeps the sub-solver transparent and reproducible within the per-epoch CPU budget.

---

**Algorithm 2** ALNS-lite for static DAPDP (time-limited)

---

**Require:** Instance $(V, N, R)$; distances $d$, $d^{\text{drone}}$; windows $[e_i, \ell_i]$; endurance $E$; time limit $T_{\max}$
**Ensure:** Feasible solution $\pi$ with trucks and drone sorties; cost $C(\pi)$
 1: $\pi \leftarrow$ GREEDY-INIT() {Earliest-deadline-first with cheapest-feasible insertion incl. rendezvous}
 2: $\pi^\star \leftarrow \pi$; $C^\star \leftarrow C(\pi)$; $t \leftarrow 0$; initialize destroy/repair weights $w$
 3: **while** $t < T_{\max}$ **do**
 4:    Select destroy op $D \in \{\text{Random}, \text{Shaw}, \text{Worst}\}$ with prob. $\propto w_D$
 5:    Select repair op $R \in \{\text{CheapestIns}, \text{Regret-}k\}$ with prob. $\propto w_R$
 6:    $\pi' \leftarrow D(\pi)$ {Remove fragment(s) correlated by time-window/space}
 7:    $\pi' \leftarrow R(\pi')$ {Reinsert with feasibility: precedence, capacity, time windows, endurance $E$, launch–rendezvous}
 8:    **if** Feasible$(\pi')$ **and** $C(\pi') \leq C(\pi)$ **then**
 9:       $\pi \leftarrow \pi'$
10:    **else**
11:       Accept $\pi'$ with prob. $p = \exp\big(-(C(\pi') - C(\pi))/\tau\big)$ {Simulated annealing acceptance}
12:    **end if**
13:    **if** $C(\pi) < C^\star$ **then**
14:       $\pi^\star \leftarrow \pi$; $C^\star \leftarrow C(\pi)$; update weights $w$
15:    **end if**
16:    $t \leftarrow$ elapsed time
17: **end while**
18: **return** $\pi^\star$

---

**Notes.**

- *Shaw* destroy method uses spatial/time-window similarity to expose consolidation; *Regret-k* encourages look-ahead when placing drone-eligible nodes.
- Feasibility checks enforce pickup-before-delivery, truck load flow, drone launch i $\rightarrow$ serve j $\rightarrow$ rejoin k with $(d^{\text{drone}}ij + d^{\text{drone}}jk) \leq E$, and time propagation consistent with constraints (Equations 11–17).

### E.3 SYNTHETIC INSTANCE GENERATOR (PUBLIC RELEASE)

**Goal.** Reproducible static and dynamic DAPDP instances that match the marginal statistics of the confidential data while remaining fully shareable.

**Geometry & travel.**

- Bounding box: $10 \times 10$ km square; depot at center.
- Truck speed: 35 km/h; Drone speed: 60 km/h (straight-line).
- Truck road distances $d_{ij}$ via Euclidean $\times$ detour factor $\rho \sim \mathcal{N}^+(1.25, 0.1)$; drone distances $d_{ij}^{\text{drone}}$ Euclidean.
- Service times: $T_i \sim \text{Uniform}[2, 5]$ min (pickup and delivery nodes if split).

**Requests (pickup–delivery pairs).**

- $|R| \sim \text{Uniform}[100, 200]$; each request $(p_i, d_i)$ sampled i.i.d. in the box with minimum separation 100 m.
- Payload $w_i \sim \text{LogNormal}(\mu = 1.2, \sigma = 0.5)$ kg, truncated to $[0.2, 8]$ kg.
- Drone-eligibility: $w_i \leq 5$ kg $\wedge \|p_i - d_i\| \leq 7$ km.
- Time windows: draw start $s \sim \text{Uniform}[0, 300]$ min; slack $\Delta \sim \text{Triangular}(20, 60, 120)$ min; set $[e_i, \ell_i] = [s, s + \Delta]$ shifted for travel feasibility; tighten 20% of requests by halving $\Delta$ to induce urgency.
- Truck capacity $Q = 200$ parcels; drone endurance $E \in \{15, 20, 25\}$ min per scenario.

**Dynamic reveal.**

- Episodes have $T \in \{5, \ldots, 9\}$ epochs; at epoch $t$, reveal $n_t \sim \text{Poisson}(18)$ then cap to keep $\leq 100$ active.
- A request is *must-go* if $\ell_i$ would be violated by deferring one more epoch.
- Feature fields precomputed for RL: drone-eligibility flag; endurance slack $\left(E - (d_{ij}^{\text{drone}} + d_{jk}^{\text{drone}})\right.$ for best $k\big)$; 5-NN density; LoS/road detour ratio $\min_k (d_{ik} + d_{kj})/\|i - j\|$.

**Objective & costs.**

- Minimize truck distance $+ \alpha \cdot$ drone distance with $\alpha = 0.3$ (default).
- Penalties follow the structured schedule below (as discussed in Section D.4) with scale $\beta \in [0.5, 1.5] \times 2.0$.

