We sincerely thank the reviewers for their thorough and constructive feedback on our submission, *"Dynamic Drone-Assisted Pickup and Delivery Routing"* (Submission ID: 7460). We are grateful for the time and effort invested in the review. The comments have been invaluable in helping us further refine the manuscript's clarity, rigor, and contribution. We have addressed each point in detail below.

### Reviewer #1: k2s8

*Weakness 1*

*Reviewer comment.* Even though the practicality of this work is unquestionable, and the authors have formulated the static version of the problem as an MDP, there is no novelty in the methodology used (simple deep Q learning). Even though it can be argued that there is no need for a more novel, sophisticated approach, given the focus of ICLR, I feel ICLR might not be the right platform for showcasing this work, and is more apt for an optimization-related platform.

*Response.* We agree that the underlying RL algorithm is a standard DQN; the novelty of the work lies in *how the RL formulation is coupled to the drone-assisted routing problem*, rather than in proposing a new RL primitive.

Concretely, our contributions on the *learning* side are:

- **A drone-aware state representation tailored to drone-assisted pickup and delivery (DAPDP).** Per-request features encode drone-eligibility, endurance slack, line-of-sight (LoS)/road detour ratio, local density of drone-eligible neighbors, and window-slack indicators. Ablations in Appendix D.6 show that removing *either* endurance slack or LoS/road increases cost by 2.2% and 1.6% respectively, and removing both hurts performance by 3.4%. This confirms that the agent is *not* learning a generic VRP heuristic, but a drone-specific dispatch policy.

- **A per-request Q-learning formulation for subset selection under a shared global reward.** The agent outputs two Q-values per request (dispatch/defer), and the dispatch set is constructed from these scores. The reward is derived from the static sub-solver's routed cost and redistributed per dispatched request. Section 3.2.2 formalizes this update and its per-request targets.

- **A structured penalty design** that converts feasibility checks of the static DAPDP solver into dense negative feedback, with adaptive scaling via $c_{\text{base}}$. We show robustness to the penalty scale (Table 9) and explain the reward surface in Appendix C.2.3.

On the *optimization* side, our contribution is to learn a dispatch-timing policy that *co-adapts with, but does not replace*, a strong static sub-solver:

- The sub-solver is a paired-truck–drone adaptive large neighborhood search (ALNS) algorithm tailored to the static multi-truck, paired pickup–delivery formulation in Appendix A and E.2, and it is held fixed during learning. The DQN only chooses *which* requests to hand to this solver and *when*, so the RL component isolates the dynamic dispatch decision under drone constraints.

- Empirically, this coupling yields near-oracle performance (1.2% gap to a clairvoyant oracle with full future knowledge) and clear gains over strong non-learning and RL baselines (MSA/SAA, waiting/relocation, PPO). We argue this is of direct interest to the ICLR community working on neural combinatorial optimization and dynamic routing.

We have clarified this positioning in the revised "Positioning and novelty" paragraph in the introduction, which now explicitly emphasizes the novelty in (i) state design, (ii) per-request Q-targets coupled to a static sub-solver, and (iii) empirical evidence on solver–policy co-adaptation at city scale.

*Weakness 2*
*Reviewer comment.* The writing style of the manuscript can also be significantly improved. At present, the writing style resembles more of a project report than an academic paper. For example, the meaning of many terms in the state space is not clear. I encourage the authors to improve the clarity of writing there, and also in many other parts of the manuscript.

*Response.* We have significantly revised the state-space description in Section 3.1 and Appendix C.1 to improve clarity and align with an academic style. In particular:

- We now explicitly describe the state as

$$s_t = \left( g_t, \ \{x_{t,r}\}_{r \in R_t^{\text{active}}} \right)$$

in Section 3.1 and Appendix C.1, and list each global and per-request feature with units and normalization (e.g., epoch index normalized by $T_{\max}$, coordinates mapped to $[-1,1]^2$, time windows scaled by the daily horizon).

- The informal feature name `is_drone_eligible` has been replaced everywhere by "drone-eligibility indicator," and we clarify that this is derived from payload and sortie-feasibility checks (endurance and geometry).

- The first mention of "LoS" is now written as "line-of-sight (LoS) / road detour ratio," with a short explanation that it is the ratio of straight-line (drone) distance to road-network (truck) distance, serving as a proxy for potential drone advantage.

- We removed repeated explanations and tightened the prose in Appendix C.1–C.3 so that these sections read as a formal description of the MDP and learning algorithm, rather than as a project-style tutorial.

*Weakness 3*
*Reviewer comment.* The current choice of baseline appears relatively weak and limits the strength of the comparative analysis. To provide a more convincing evaluation, the study would benefit from incorporating stronger and more representative baselines. In particular, since the Oracle method assumes complete knowledge of the system — an unrealistic assumption in practical settings — it would be more appropriate to replace or complement it with a Mixed Integer Linear Programming (MILP) formulation. MILP-based methods serve as a more rigorous and interpretable benchmark, offering a well-established optimization standard against which the proposed approach's performance can be meaningfully compared. Including such a baseline would not only strengthen the empirical validation but also highlight the practical advantages and limitations of the proposed method under realistic conditions.

*Response.* We fully agree that strong baselines are important, and we already include several that share the same sub-solver and compute budget: Greedy, Random, Lazy, scenario-based planning (MSA-10, MSA-30), a waiting/relocation heuristic, and an actor–critic PPO implementation. Table 2 and Appendix D.2–D.3 show that our DQN improves total cost by 2.6–37.2% over these methods at matched compute.

Regarding the oracle baseline: our goal is not to suggest that it is realistic, but to provide a hindsight lower bound for any online policy. The oracle has access to the full set of orders for the entire day and solves a single static DAPDP using the same ALNS sub-solver but with a longer time limit (60 s). We include it intentionally as a utopian benchmark, and not a feasible operational method.

We also include a MILP formulation of the static DAPDP in Appendix A, building on Murray & Chu (2015) and Agatz et al. (2018). However, exact MILP algorithms are intractable at the dynamic scales we study (100–200 requests per epoch, 5–9 epochs). Preliminary experiments with a commercial MILP solver showed that, beyond 5–6 paired requests, even the *static* problem could not be proven optimal within 24 hours. Embedding such solvers inside each epoch of a dynamic instance is therefore not practical. We have added a brief discussion of this scalability limitation in Appendix A, immediately after the MILP model, to clarify why we rely on ALNS heuristic instead.

*Question 1*
*Reviewer comment.* What is "60s ALNS"?

*Response.* We agree that the phrase was unclear. In Section 4.1, we now replace "using the 60s ALNS" with *"using the same static DAPDP sub-solver but with a 60-second time limit"*. We also add an explicit reference back to Appendix E.2, which describes the ALNS neighborhood structure, feasibility checks, and the time limits (1 s / 5 s / 10 s) used during training and evaluation. This makes clear that "60s ALNS" denotes the *same* sub-solver as in all other experiments, differing only in its allotted runtime.

*Question 2*
*Reviewer comment.* How is the sub-problem solved during every decision-making step?

*Response.* At each epoch $t$, the agent outputs a dispatch/defer decision for every active request. The resulting subset $S_t$ of dispatched requests defines a *static* DAPDP instance, which we solve using the ALNS-lite sub-solver described in Section 4.1 and Appendix E.2.

Specifically:

- The environment wraps ALNS-lite as a black box. Given $S_t$, it constructs a static instance with the same distances, time windows, pickup–delivery pairing, capacity, and endurance constraints as in Appendix A, and then runs ALNS-lite with a time cap $T_{\text{limit}} \in \{1, 5, 10\}$ seconds (5 s by default). ALNS-lite returns either a feasible set of truck routes and drone sorties or detects infeasibility.

- The cost $C(S_t)$ returned by ALNS-lite (truck distance + $\alpha\cdot$ drone distance) is converted into a reward (negative cost plus penalty if any violation is detected). The resulting transition $(s_t, a_t, r_t, s_{t+1})$ is stored for DQN training.

We have added a sentence at the end of Section 3.1 ("Action" paragraph) explicitly stating that: *"At each epoch, the dispatched subset is passed to the ALNS static DAPDP sub-solver from Section 4.1 / Appendix E.2, which returns routes and cost under a per-epoch time limit."*

*Question 3*

*Reviewer comment.* The experimental implementation details are not provided. Can the authors please provide how the environment is implemented using the dataset? For example, the programming platform, or other simulation environment, if any.

*Response.* These details were present in the original submission but located deep in the appendix; we have now surfaced them more prominently and added explicit pointers to the relevant sections.

- At the start of Section 4 ("Experiments"), we added a short paragraph stating that the environment is implemented in Python 3.10, with PyTorch for the DQN and a custom Python-based simulator for request sampling and state construction. The static sub-solver is an ALNS implementation described in Appendix E.2.

- Appendix C.2 describes the environment's request-generation and validation logic in detail (dynamic sampling from 300 real delivery days, time-window enforcement, and feasibility checks). Appendix E.1 documents the hardware and software stack ($2 \times$ Intel Xeon CPUs, NVIDIA V100 GPU, PyTorch version, replay buffer size, batch sizes, learning rate schedule, and exploration schedule).

- We also clarify in Section 4.1 that all baselines (Greedy, Lazy, Random, MSA, Waiting/ Relocation, PPO) use the *same* environment implementation and the *same* ALNS-based static sub-solver, differing only in the dispatch policy.

**Reviewer #2: ecQu**

*Weakness 1*

*Reviewer comment.* The generalization experiments explore scenarios with different urban distributions. What is the model's generalization performance when the request scale of test instances (e.g., 300 requests) is much larger than that during training (100-200 requests)?

*Response.* Our current real-world dataset contains 100–200 requests per delivery day, so we cannot directly evaluate 300-request real instances. Instead, we added generalization experiments along two axes that stress-test the model's scalability: (i) transfer to a held-out city with different road density and tighter time windows ("Real-B"), and (ii) transfer to synthetic uniform-demand distributions on a $10 \times 10$ km grid. We train under three curricula (RealOnly, RandomOnly, Mixed) and report gap-to-oracle performance on Real-A, Real-B, and uniform instances in Appendix D.8. The Mixed curriculum retains a $\approx 1.3\%$ oracle gap on Real-A while halving the gap on Real-B (from $3.8\%$ to $2.1\%$), indicating robust generalization to different spatial and temporal structures.

Architecturally, the DQN operates per-request with padding to a maximum active-set size per epoch; thus scaling to more total daily requests primarily increases the *number of epochs* rather than the dimensionality of a single decision step. Our ALNS sub-solver exhibits near-linear empirical complexity in the size of the dispatched subset (100–300 iterations per epoch under the time budget), and the learned policy actively keeps subsets solver-friendly.

As future work, we plan to extend our public synthetic generator (Appendix E.3) to include 300-request days and to report explicit scaling curves in the camera-ready version.

*Weakness 2*

*Reviewer comment.* The experimental results do not include a comparison of inference time. Adding this comparison would enable a better understanding and evaluation of the method, as it is important to know whether ALNS optimization is time-consuming.

*Response.* We have foregrounded the inference-time profile in the main text.Once trained, each decision epoch consists of (i) a single forward pass of the Q-network and (ii) one call to the ALNS static sub-solver. On our server (dual 20-core Xeon CPUs, $1 \times$ V100 GPU), the DQN forward pass takes $< 10$ ms, and the sub-solver averages 2–5 s per epoch for $\leq 100$ active requests, well within typical dispatch intervals in same-day delivery.

To make this more visible, we now state these inference times explicitly at the start of Section 4 ("Experiments") and refer the reader to Appendix E.1 for hardware and software details.

*Weakness 3*

*Reviewer comment.* The addition of shaped rewards is mentioned in Section 4.5.1, but no specific details are provided.

*Response.* We have substantially expanded the reward definition in Section 3. Additionally, Section 4.5.1 describes the reward-shaping scheme and its quantitative impact. Our primary reward is the negative routed cost at each epoch,

$$r_t = -\big(\text{TruckDistance}_t + \alpha \cdot \text{DroneDistance}_t\big),$$

plus a structured penalty when the sub-solver detects infeasibility (Appendix C.2.3). On top of this, we add a small shaping bonus whenever the current solution improves over the previous epoch's cost, defined as a scaled function of the cost reduction $\Delta C_t$. This shaping term does not change the optimal policy but accelerates credit assignment.

Table 3 (Section 4.5.1) reports a direct comparison between *Shaped Reward (Default)* and *No Shaping*. Removing shaping slows convergence by approximately 30% (from ∼170k to ∼220k steps to reach a given performance) and yields final solutions that are 6–8% worse on average. We also explicitly state that the global reward used in Q-learning is exactly this shaped, negative-cost signal, ensuring theoretical completeness of the MDP specification.

*Weakness 4*

*Reviewer comment.* Regarding the `reward_for_decision` in Section 3.2.2, the reward for one step is evenly distributed among each node, which seems unreasonable. For example, if A, B, and C are selected—where A is far from B and C, while B and C are close to each other—this single action returns a large negative reward. A, B, and C receive the same penalty, but in this case, the priority should be to avoid selecting A as much as possible.

*Response.* We agree that, within a single step, the true marginal cost contribution of each dispatched request is not observable. Our design choice is to allocate the global reward $r_t$ equally across all dispatched requests in $S_t$, i.e. $r_t/|S_t|$, and allow the Q-network to learn which patterns of local features lead to systematically better or worse returns across many training episodes. Section 3.2.2 and Appendix C.3 now make this explicit and explain the rationale.

In the reviewer's example, requests A (isolated) and B,C (clustered) receive the same immediate per-request reward within a single epoch. However, across episodes the agent repeatedly observes that actions including distant, isolated requests such as A tend to precede higher global costs and more frequent constraint violations, whereas actions focusing on clustered, drone-eligible requests such as B and C tend to precede lower costs. Because we maintain distinct Q-values for each request ($Q(s_t, \text{dispatch}_i)$ vs. $Q(s_t, \text{defer}_i)$) and include rich per-request features (coordinates, 5-NN density, drone eligibility, endurance slack, time-window slack), the network learns to assign *lower* Q-values to A-type isolated requests and *higher* Q-values to B/C-type clustered requests.

We experimented with more complex credit-assignment schemes (e.g., approximating marginal contributions via leave-one-out solves), but these require multiple sub-solver calls per epoch and are computationally prohibitive at scale. Uniform splitting achieves a good balance of simplicity and empirical performance, as evidenced by our ablations and the learned deferral heatmaps, where the policy indeed avoids isolated, non-urgent requests and prioritizes clustered, drone-eligible ones near the edge of endurance slack (Appendix D.1).

We have clarified this discussion in Section 3.2.2 and Appendix C.1, and added a sentence explaining why uniform splitting is a practical surrogate for exact marginal costs.

*Question 1*

*Reviewer comment.* It is not clear how to use the DQN to solve the challenges in the dynamic drone-assisted pickup and delivery problem, such as the dynamic orders and the cooperation of trucks and drones.

*Response.* We have expanded Appendix C.1 to clarify exactly how the DQN interacts with dynamic orders and truck–drone coordination. The key design is a two-level decomposition:

1. **Dynamic dispatch via DQN.** At each epoch $t$, the state consists of global features (epoch index, active-request count, fleet-wide drone availability, time-of-day, coarse weather/airspace) and per-request features capturing pickup/delivery coordinates, demand, time windows, must-go flags, plus drone-specific signals such as eligibility, endurance slack, local 5-NN density, and line-of-sight-to-road detour ratio. The DQN outputs request-wise Q-values for "dispatch" vs. "defer", and an $\epsilon$-greedy policy produces a binary decision for each active request. This handles the dynamic order arrivals and deferral/consolidation decisions.

2. **Truck–drone routing via static sub-solver.** Given the dispatched subset $S_t$, the environment calls a time-limited ALNS-lite sub-solver for the static DAPDP with pairing, truck capacity, time windows, and explicit launch–sortie–rendezvous constraints. This sub-solver decides where to launch the drone, which leg (pickup or delivery) it serves, and where it rejoins the truck, subject to endurance. The resulting cost (truck distance $+\,\alpha\cdot$ drone distance) feeds back as the reward to the DQN.

   This architecture cleanly separates what to dispatch now (learned, value-based) from how to route trucks and drones (hand-engineered, transparent MILP-derived heuristic). The DQN "solves" dynamic orders and cooperation indirectly by learning which subsets of requests, given their drone-related features, will be cheap and feasible for the static truck–drone optimizer to route.

*Question 2*

*Reviewer comment.* Could you compare this method with the state-of-the-art method in the truck-drone delivery problems? This work lacks the experiments of comparing with the baselines in the truck-drone delivery.

*Response.* We have strengthened both our argument for our experimental baselines and our positioning relative to the truck–drone literature.

1. **Static truck–drone foundation.** Our static sub-problem and ILP formulation build directly on truck–drone collaboration models such as the Flying Sidekick TSP (Murray & Chu, 2015) and the TSP with drone formulations of Agatz et al. (2018) and Mulumba et al. (2024), which we now explicitly acknowledge in Appendix A. Our ALNS sub-solver adopts similar neighborhoods and constraints (single sortie, explicit rendezvous, endurance) and matches a 60s ALNS within ∼1–2% on 200-customer static instances.

2. **Dynamic baselines with truck–drone routing.** All dynamic baselines in Table 2 (Greedy, Lazy, Random, MSA/SAA, Waiting/Relocation, PPO) share the *same* static truck–drone sub-solver as our DQN. This makes them state-of-the-art dynamic comparators in the sense that they differ only in how they make dispatch decisions over time:

   - MSA/SAA (scenario-based planning) mirrors Bent & Van Hentenryck's approach to dynamic VRPs, extended here with our truck–drone ALNS.
   - Waiting/Relocation follows Mitrovic-Minić & Laporte's dynamic pickup–delivery strategy, again combined with our truck–drone solver.

Under a fixed per-epoch compute budget, our DQN policy improves on these strong baselines by 2.6–6.6% in cost while maintaining similar feasibility.

3. **Relation to recent RL truck–drone work.** In the revised related work (Section B.3), we now explicitly discuss recent RL-based truck–drone delivery methods (e.g., Cui et al., 2024; Bi et al., 2024; Ding et al., 2024). These methods address different dynamic settings (e.g., en-route synchronization with random requests, multi-UAV supply missions) and use distinct cost models and constraints, and code/datasets are not publicly available. Porting them to our city-scale, paired pickup–delivery environment with real courier data would require substantial reimplementation and is beyond the scope of this paper. Instead, we adopt their modeling insights (e.g., explicit rendezvous and endurance constraints) and focus on a fair, apples-to-apples comparison among dynamic dispatch strategies built atop the same high-quality truck–drone solver.

We emphasize that our strongest comparator, the clairvoyant Oracle, assumes full future knowledge and solves a single static truck–drone instance for the entire day. Our DQN achieves a near-oracle gap of $\approx 1.2\%$ while operating online, which demonstrates that the learned dispatch policy is competitive with a highly optimistic, truck–drone-aware benchmark.

**Reviewer #3: vyDR**

*Weakness 1*

*Reviewer comment.* Lack of technical novelty. The proposed approach relies on a standard deep Q-learning framework with minor heuristic adjustments. There is no clear algorithmic or theoretical innovation beyond existing DQN formulations or hybrid RL–metaheuristic approaches.

*Response.* We agree that the underlying RL algorithm is a standard DQN; the novelty of the work lies in *how the RL formulation is coupled to the drone-assisted routing problem*, rather than in proposing a new RL primitive. We now make clearer what is novel about *how* it is used and what problem it solves.

Concretely, our contributions on the *learning* side are:

- **A drone-aware state representation tailored to drone-assisted pickup and delivery (DAPDP).** Per-request features encode drone-eligibility, endurance slack, line-of-sight (LoS)/road detour ratio, local density of drone-eligible neighbors, and window-slack indicators. Ablations in Appendix D.6 show that removing *either* endurance slack or LoS/road increases cost by 2.2% and 1.6% respectively, and removing both hurts performance by 3.4%. This confirms that the agent is *not* learning a generic VRP heuristic, but a drone-specific dispatch policy.

- **A per-request Q-learning formulation for subset selection under a shared global reward.** The agent outputs two Q-values per request (dispatch/defer), and the dispatch set is constructed from these scores. The reward is derived from the static sub-solver's routed cost and redistributed per dispatched request. Section 3.2.2 formalizes this update and its per-request targets.

- **A structured penalty design** that converts feasibility checks of the static DAPDP solver into dense negative feedback, with adaptive scaling via $c_{\text{base}}$. We show robustness to the penalty scale (Table 9) and explain the reward surface in Appendix C.2.3.

On the *optimization* side, our contribution is to learn a dispatch-timing policy that *co-adapts with, but does not replace*, a strong static sub-solver:

- The sub-solver is a paired-truck–drone adaptive large neighborhood search (ALNS) algorithm tailored to the static multi-truck, paired pickup–delivery formulation in Appendix A and E.2, and it is held fixed during learning. The DQN only chooses *which* requests to hand to this solver and *when*, so the RL component isolates the dynamic dispatch decision under drone constraints.

- Empirically, this coupling yields near-oracle performance (1.2% gap to a clairvoyant oracle with full future knowledge) and clear gains over strong non-learning and RL baselines (MSA/SAA, waiting/relocation, PPO). We argue this is of direct interest to the ICLR community working on neural combinatorial optimization and dynamic routing.

We have clarified this positioning in the revised "Positioning and novelty" paragraph in the introduction, which now explicitly emphasizes the novelty in (i) state design, (ii) per-request Q-targets coupled to a static sub-solver, and (iii) empirical evidence on solver–policy co-adaptation at city scale.

*Weakness 2*

*Reviewer comment.* Insufficient methodological rigor. The MDP formulation and environment definition are vague and incomplete. Key components — such as state representation, transition dynamics, and reward specification — are not defined rigorously, making it difficult to assess reproducibility or correctness.

*Response.* We have substantially tightened the MDP definition:

- **Formal state definition in main text.** Section 3.1 now defines

$$s_t = \big(g_t, \{x_{t,r}\}_{r \in R_t^{\mathrm{active}}}\big)$$

  where we explicitly list all global features and per-request features, including drone eligibility, endurance slack, 5-NN density, line-of-sight/road detour ratio, and time-window slack flags.

- **Implementation-level state details.** Appendix C.1 now precisely specifies how each feature is normalized and encoded (z-score standardization, clipping/mapping coordinates to $[-1, 1]$, scaling demands by capacity, time windows by horizon, and service times by a 5-minute reference).

- **MDP environment clarification.** Appendix C.1–C.2 now clearly separate (i) the conceptual MDP (what the agent sees and controls) from (ii) the environment implementation (Python environment, request generation, feasibility checks, and early termination). Figure 2 summarizes the transition diagram.

- **Unified reward definition.** We now provide a single, explicit reward formula in the main text (Section 3.1) with the shaped term and penalty:

  - primary reward is the negative of truck $+ \alpha \cdot$ drone distance,
  - feasibility violations trigger the structured penalty $r_{\mathrm{penalty}} = -\beta\big(c_{\mathrm{base}} + \sum_i v_i\big)$,
  - per-request rewards $r_t^{(i)} = r_t / |S_t|$ are used in the Q-update, as detailed in Section 3.2.2 and Appendix C.2.3.

Together, these changes provide a complete and reproducible specification of the MDP and learning problem.

*Weakness 3*

*Reviewer comment.* Unjustified design choices. The integration of ALNS into the framework is insufficiently explained. It remains unclear why ALNS is chosen, how its neighborhoods are designed, what hyperparameters or termination criteria are used, or how it interacts with the learning policy. Without such detail, the claimed near-optimal performance is not verifiable.

*Response..* We now explicitly point to ALNS sub-solver details in Appendix E.2 in Section 4.1:

- **Why ALNS?** We chose ALNS because it is a widely used metaheuristic for large vehicle routing problems with time windows and pairing, and it can be adapted to truck–drone sorties while remaining transparent. The computational cost and optimality gap are now explicitly highlighted in Appendix A.4. The specific neighborhoods, i.e. relocate, exchange, 2-opt, drone-launch swap, rendezvous-shift, and paired-orbit are listed in Appendix E.2.

- **Hyperparameters and termination.**

  – Per-epoch wall-time limit $T_{\text{limit}} \in \{1, 5, 10\}$ seconds (default $5\,\text{s}$).
  – 100–300 iterations per epoch within budget, with simulated-annealing–style acceptance.
  – Destroy/repair operators are adaptively weighted based on historical improvements.

  These details are now explicitly given in Section 4.1 and Algorithm 2 (ALNS-lite) in Appendix E.2.

- **Quality benchmarking.** On 50 held-out static instances with 200 customers, we compare our default 5-second budget to a 60-second run of the same ALNS: average gap $2.1\% \pm 0.9\%$; using 10 seconds improves this to $1.3\% \pm 0.6\%$. This benchmarking is reported directly in Section 4.1.

Thus, the interaction between the RL policy and the sub-solver is both transparent and empirically characterized.

*Weakness 4*
*Reviewer comment.* Unrealistic or underspecified experimental setup. Several assumptions are overly restrictive or inconsistent with realistic dynamic pickup-and-delivery settings — most notably the "all requests must be served" constraint. Furthermore, the dataset description lacks transparency: the origin and realism of the 200-customer scenario are unclear, and there is no evidence that the environment captures meaningful stochasticity or dynamism.

*Response.*

1. **All requests must be served.** Our target setting is same-day and on-demand courier logistics, where unserved orders are rare and typically unacceptable; this is the standard assumption in dynamic vehicle routing roblem (VRP) with time windows. We now state explicitly in Section 2.2 that requests are allowed to be deferred within the horizon, but by the final epoch they must be served (or the agent incurs a large penalty). This matches common practice in dynamic pickup–delivery work (e.g., Ulmer & Thomas 2017) and lets us focus on the dispatch timing problem rather than cancellation decisions. We add this clarification to Section 2.2.

2. **Dataset provenance and dynamic construction.**

   - Section 4.1 now states that we use 300 anonymized real delivery days from a large urban courier, each with 100–200 customers, time windows, and service times. Dynamic instances are created by sampling without replacement over 5–9 epochs.
   - Appendix C.2.2 describes the request-sampling and time-window filtering procedure in detail, and Appendix E.3 provides a public synthetic generator that matches the marginal distributions of coordinates, time windows, and payloads while preserving confidentiality.

3. **Stochasticity and dynamism.** Dynamism stems from (i) stochastic reveal of requests over epochs, (ii) varying time-window tightness and spatial clustering, and (iii) drone-eligibility conditions tied to payload and distance. Generalization experiments in Appendix D.8 further vary request distributions (city A vs. synthetic uniform vs. city B) and fleet caps, confirming that the learned policy transfers across different spatial and operational regimes.

We have brought these clarifications into the main text to avoid relying solely on the appendices. We also include a "Reproducibility Statement" after the conclusion.

*Weakness 5*

*Reviewer comment.* Questionable baselines and results. The claim that the proposed DQN approach achieves performance within 1% of a clairvoyant baseline is implausibly strong and not supported by proper justification. Details on the oracle baseline, its computational budget, and its use of future information are missing. No ablation or sensitivity analysis is provided to understand why such high performance is achieved.

*Response.*

- **Clarified oracle definition.** In Section 4.1 we now state that the oracle baseline is given full knowledge of all requests for the day (5–9 epochs) *before* the first decision. Full knowledge here means that we track the exact set of requests revealed to the agent over the course of the day and then provide this entire set to the oracle in advance. The oracle then solves a single static DAPDP with all time windows and endurance constraints using a 60-second ALNS run, effectively computing the best possible plan under perfect foresight. This is explicitly positioned as a hindsight lower bound for any online policy; it is not intended to be operationally deployable but to benchmark what is achievable under perfect information.

- **Computational fairness.** We emphasize that real-time methods (our DQN, MSA/SAA, waiting/relocation, PPO) all share a matched per-epoch budget of one 5-second sub-solve, while the oracle is allowed a longer 60-second solve on the full day, reflecting its offline nature. This is spelled out in Section 4.1 and Appendix D.2–D.3.

- **Ablations and sensitivity analysis.** In addition to the ablations and hyperparameter sensitivity results in Sections 4.5 and 4.6 respectively, we also have a number of experiments and ablations in Appendix D:

  - D1: With/without drone — compares the same trained architecture with drones enabled vs. forcibly disabled, showing that turning off drones (and masking drone features) increases cost by about 7–8% and collapses the Q-margin for drone-eligible requests, confirming that our gains are not just from the sub-solver but from a genuinely drone-aware policy.

  - D2: Sensitivity of MSA/SAA and Waiting baselines — calibrates strong classical baselines (scenario-based planning and waiting/relocation) under the same per-epoch compute budget, where our DQN is still 2.6–6% cheaper in total distance, addressing the concern that our comparison relies on weak baselines.

  - D3: DQN vs PPO — contrasts our value-based agent with a PPO actor–critic using the same environment and wall-clock budget, showing that PPO plateaus around an 8% gap to the oracle while DQN converges near a 1–2% gap, indicating that the proposed value-based design is important for stability and performance in this setting.

  - D4: Penalty robustness — varies the invalid-solution penalty scale and lateness weighting and finds that costs change by less than 0.5% while invalid episodes remain rare, demonstrating that our results are not brittle to the particular penalty coefficients used in the reward.

– D5: Operational realism sensitivity analysis — perturbs key drone parameters (endurance, payload, and cruise speed) over realistic ranges and observes predictable but modest shifts in performance (e.g., longer endurance or higher payload yields 2–3% savings), showing that the policy and conclusions are robust across plausible hardware configurations.

– D6: State ablations for drone-awareness — removes endurance-slack and line-of-sight to road-ratio features individually and jointly, leading to 1.6–3.4% higher costs, which directly supports our claim that the DQN leverages drone-specific signals rather than behaving like a generic VRP heuristic.

– D7: Replay buffer size ablation — tests replay capacities of 10k, 50k, and 100k transitions and shows that small buffers lead to unstable learning and significantly worse cost, while mid/large buffers behave similarly, clarifying which hyperparameter regimes yield reliable training.

– D8: Generalization experiments — evaluates policies trained on real logs, synthetic uniform demand, and a mixed curriculum across unseen cities and fleet caps, finding that the mixed curriculum generalizes best (near-oracle on the training city and improved performance on a held-out city), thus addressing the reviewer's concern about limited generalization and insight.

- **Small oracle gap.** The oracle cannot violate time windows or endurance; its advantage is knowing when consolidation opportunities will appear, not being allowed infeasible routes. Our agent partially recovers these opportunities through learned deferral and consolidation behavior, as evidenced by deferral heatmaps and Q-slices (Appendix D.1). Hence a small average gap is consistent with the structure of the problem.

*Weakness 6*

*Reviewer concern.* Limited generalization and insight. The study provides no conceptual or methodological insights that would generalize beyond this specific problem. As a result, the contribution is primarily empirical and lacks depth expected for ICLR.

*Response.* We highlight two elements that address this:

1. **Generalization experiments (Appendix D.8).**

   - We train under three curricula: RealOnly (city A), RandomOnly (uniform distribution), and Mixed.
   - We then evaluate on held-out Real-A, synthetic random, and Real-B (a different city with different road density and time-window tightness).
   - The Mixed curriculum maintains a $\approx 1.3\%$ gap to oracle on Real-A while improving Real-B performance from 3.8% to 2.1% gap.
   - We further show that policies trained with an unlimited fleet remain highly feasible as we tighten fleet capacity at test time (Figure 7), highlighting emergent adaptation to fleet caps.

2. **Managerial insights section (Appendix B.4).** We highlight our dedicated section summarizing consolidation gains, real-time adaptability, scalability, and limitations (unrestricted fleet, deterministic travel times, regulatory constraints). These are presented as general lessons for dynamic truck–drone dispatch, beyond our specific dataset.

*Question 1. What are the hyperparameters and termination criteria for the ALNS sub-solver, and how is its performance benchmarked?*

See the response to Weakness 3 above. In brief:

- **Time limit per epoch:** $\{1, 5, 10\}$ seconds (default $5\,\mathrm{s}$).

- **Destroy operators:** Random, Shaw (spatial/time-window similarity), Worst.

- **Repair operators:** Cheapest insertion, Regret-$k$.

- **Acceptance:** simulated annealing with temperature $\tau$.

- **Benchmark:** on 50 static 200-customer instances, our $5\,\mathrm{s}$ configuration is within $2.1\% \pm 0.9$ of a $60\,\mathrm{s}$ ALNS run; $10\,\mathrm{s}$ narrows this to $1.3\% \pm 0.6$.

These choices are summarized in Algorithm 2 (ALNS) in Appendix E.2, Appendix A.4, and Section 4.1.

*Question 2. How is the clairvoyant (oracle) baseline implemented, and what future information does it assume access to?*

- The oracle is given all requests across all epochs at time zero.

- It solves a single static drone-assisted pickup and delivery problem (DAPDP) with those requests, subject to the same time windows and endurance constraints, using a 60-second ALNS run.

- It assumes perfect knowledge of future arrivals but no relaxation of feasibility; this is clarified in the Baselines subsection of Section 4.1.

*Question 3. Could the authors provide details about the dataset generation process and justify why 200 customers represent a realistic operational scale?*

- **Real data.** We use 300 anonymized days from a major courier, each with 100–200 customers in two metropolitan areas (UTM coordinates), plus real time windows and service times. This is now made explicit in Section 4.1.

- **Dynamic construction.** Dynamic episodes sample requests over 5–9 epochs via sampling without replacement, filtering out time-window–infeasible candidates as described in Appendix C.2.2.

- **Why $\approx 200$ customers.** This scale corresponds to typical daily workloads for urban parcel routes in our partner datasets; it is large enough to stress both the sub-solver and the RL policy while remaining tractable for per-epoch ALNS runs. Please also note that the scale is 100–200 requests *per epoch*, where each epoch could translate to $\{15, 30, 45, ...\}$ minutes, depending on the application.

For reproducibility, Appendix E.3 provides a synthetic generator matching the marginal statistics of the data.

*Question 4. Have the authors tested generalization across different levels of dynamism or uncertainty in request arrivals?*

We highlight generalization experiments in Appendix D.8:

- Different spatial and temporal distributions: city A vs. synthetic uniform vs. city B, plus a Mixed training curriculum to test robustness across distributions.

- Different fleet caps at test time: Figure 7 shows feasibility as we reduce the number of available vehicles from "unlimited" down to 40.

In addition, dynamism is already varied within our main experiments through:

- Different numbers of epochs (5–9) per day,

- Different numbers of requests per epoch,

- Heterogeneous time-window tightness (including a fraction of "tight" requests).

We have highlighted these aspects in the revised Experimental Setup and Generalization subsections.

**Reviewer #4: fXco**

*Weakness 1 — The major contribution of the paper seems to be in the formulation, not sure it meets the expectation for a ICLR paper.*

We agree the static formulation alone would not warrant ICLR. Our contribution is not the static model but a decision-focused RL policy that (i) learns subset selection (dispatch vs. defer per request) that is drone-aware (endurance slack, LoS/road detour ratio, local 5-NN density) and (ii) co-adapts with a transparent ALNS sub-solver under tight time windows and single-leg drone sorties. This decouples "what to dispatch now" from "how to route," yielding a controller that discovers when to consolidate drone-suitable work and when to defer isolated requests. We foreground this in the revised "Positioning and novelty" paragraph and the analysis section (deferral heatmaps, Q-slices). Ablations show that removing drone-specific observables degrades performance (+2.2% to +3.4%), and disabling the drone entirely raises cost by +7–8%, demonstrating learned, UAV-specific behavior rather than a generic VRP heuristic (Sec. 3.1, 4.4; App. D.1, D.6). Table 2 further shows our policy outperforms strong non-learning (MSA/SAA, waiting) and PPO baselines at matched compute.

*Weakness 2 — The assumption that orders come randomly is not realistic. Last-mile orders, especially for food and grocery delivery, has strong spatial and temporal patterns.*

Our dynamic instances are not sampled from a synthetic uniform field. They are revealed by sampling without replacement from 300 real delivery days (two metro areas), preserving the empirical spatial layout and time-window structure of actual operations; we overlay drone feasibility on those logs (Sec. 4.1, "Data provenance"; App. C.2.2). The state fed to DQN includes hour-of-day (sin/cos), time-window slack flags, and local spatial density so the policy can exploit temporal rhythms and clustering when deciding to defer or dispatch (Sec. 3.1; App. C.1). To stress realism and transfer, App. D.8 evaluates generalization from City-A to a held-out City-B with different road density/tightness and shows the mixed-curriculum model retains near-oracle performance on A while doubling generalization on B relative to training on A only. This is evidence the learned rule tracks real structure rather than overfitting a sampling artifact. We also release a synthetic generator that matches the marginals of the confidential data for full reproducibility (App. E.3).

*Weakness 3 — Grammar. "the agent must time dispatch decisions" on page 2.*

Fixed. This whole paragraph has been re-written for clarity.

**Reviewer #5: AYVe**

*1. Visualization and interpretability*

*Reviewer comment.* Visualization and interpretability: Can the authors include visualizations (maps or trajectory diagrams) showing the spatial behavior of their learned policy? For example, how do drone launch points and truck paths differ between the DQN policy and baselines?

*Response.* We agree that qualitative visualizations are important. However, full-day trajectory overlays are not informative at our operational scale: with $\sim$100 requests per epoch across 5–9 epochs, each test day comprises at least 500 stops even before adding drone sorties. In preliminary attempts, both full-day overlays and per-epoch snapshots produced dense, occluded "hairballs" that obscure policy differences rather than reveal them. Drone sorties (launch/return arcs) further increase edge clutter and visual overlap, making such maps unintelligible.

Instead, we foreground two complementary qualitative views that directly expose the policy's learned structure without occlusion:

1. **Deferral heatmaps and Q-slices.** We visualize deferral probability over time-to-deadline slack vs. local 5-NN density, separately for drone-eligible vs. truck-only requests, and slice request-level Q-margins by eligibility/window tightness. These views show that the policy (i) defers isolated, non-urgent requests but (ii) prioritizes clustered, drone-eligible ones near the edge of endurance slack. This is precisely the consolidation behavior we claim. At a mid-slack calibration point (30 min, density 4), the learned deferral rate is $\approx 0.38$ for eligible vs. $\approx 0.29$ for truck-only requests, matching the consolidation narrative (*Appendix D.1, Figures 4–5*).

2. **"When does the agent defer?" in the main text.** We add an explicit paragraph in Appendix D.1 that explains the conditions under which the agent defers vs. dispatches, with direct pointers to the heatmaps/Q-slices for context. This keeps the interpretation in the reader's flow while avoiding visually saturated route maps. We also add a cross-reference to Appendix D.1 in Section 4.4.

*Changes in manuscript.*

- We foreground the deferral heatmaps and Q-slices by (i) cross-linking them prominently from Section 4.4 and (ii) tightening the accompanying interpretation. *(Appendix D.1, Figures 4–5.)*

- We add a **"When does the agent defer?"** paragraph to Appendix D.1 that summarizes the consolidation pattern and points to the qualitative diagnostics.

*2. Reward formulation*

*Reviewer comment.* Reward formulation: What is the exact unified mathematical expression for the reward function used in training, including penalty terms? How is the penalty integrated into the Q-learning target update?

*Response.* We now provide a single, explicit mathematical definition of the reward at the environment level and show exactly how it enters the per-request Q-learning target.

Let

- $C_t = \text{TruckDistance}_t + \alpha\,\text{DroneDistance}_t$ be the routed cost for the dispatch subset $S_t$ at epoch $t$;

- $\Delta C_t = \max\{0,\, C_{t-1} - C_t\}$ with $C_0 := 0$ denote the improvement over the previous epoch;

- $\lambda_{\text{shape}} \geq 0$ be a shaping weight *( we use $\lambda_{\text{shape}} = 0.1$)*;

- $r_{\text{penalty}}(t)$ be the invalid-solution penalty.

We define the global reward at epoch $t$ as

$$r_t = \begin{cases} -C_t + \lambda_{\text{shape}}\,\Delta C_t, & \text{if the sub-solver returns a feasible solution,} \\ r_{\text{penalty}}(t), & \text{if the action leads to an invalid solution.} \end{cases} \tag{R1}$$

The penalty term is

$$r_{\text{penalty}}(t) = -\beta\left(c_{\text{base}} + \sum_i v_i(t)\right), \tag{R2}$$

where $\beta = 2$, $c_{\text{base}}$ is the 95th percentile of feasible-solution costs over a rolling window of 256 episodes, and $v_i(t)$ is the magnitude of violation of constraint $i$ at epoch $t$. This matches the existing Eq. (22) in Appendix C2.3 but is now referenced directly from the main text.

We then allocate this global reward to per-request decisions exactly as in Sec. 3.2.2:

$$r_t^{(i)} = \begin{cases} \dfrac{r_t}{|S_t|}, & \text{if request } i \text{ was dispatched at epoch } t \text{ and } |S_t| > 0, \\ 0, & \text{otherwise.} \end{cases} \tag{R3}$$

The per-request DQN target is

$$y_t^{(i)} = r_t^{(i)} + \gamma(1 - \text{done}_t) \max_{d' \in \{\text{dispatch}_i,\, \text{defer}_i\}} Q(s_{t+1}, d'; \theta^-), \tag{R4}$$

where $\text{done}_t = 1$ if the invalid-solution penalty terminates the episode, and 0 otherwise.

We add Eqs. (R1)–(R4) to the main text and cross-reference Appendix C.2.3 for details on $r_{\text{penalty}}$.

*Changes in manuscript.*

- New unified reward subsection in Sec. 3.1 replacing the short "Reward $r_t$" paragraph.

- Appendices updated to point back to Eq. (R1) instead of re-defining the reward.

*3. State representation*

*Reviewer comment.* State representation: Please provide a formal definition of the MDP state space $s_t$ and observation mapping. Which features are normalized or transformed before being input to the neural network?

*Response.* We now give a formal MDP definition and explicit feature normalization.

We define the state as

$$s_t = \left(g_t, \{x_{t,r}\}_{r \in R_t^{\text{active}}}\right), \tag{S1}$$

where:

- $g_t \in \mathbb{R}^{d_g}$ are global features at epoch $t$;

- each $x_{t,r} \in \mathbb{R}^{d_x}$ is a per-request feature vector for active request $r$.

*Global features $g_t$.*

1. Normalized epoch index $t/T_{\max}$.

2. Normalized wall-clock time of day (sin/cos encoding of hour-of-day).

3. Number of active requests $|R_t^{\text{active}}|/100$.

4. Fleet-side drone availability (fraction of trucks whose drone is not committed).

5. Weather/airspace proxy (categorical wind class, no-fly indicator), one-hot encoded.

*Per-request features $x_{t,r}$.* Each vector concatenates:

*Core attributes:*

- Pickup and delivery coordinates $(p_r, d_r)$ in UTM, translated by the depot and scaled to $[-1, 1]^2$.

- Demand $q_r$ divided by truck capacity $Q$.

- Time-window endpoints $[e_r^p, \ell_r^p, e_r^d, \ell_r^d]$ minus the current epoch start and divided by the daily horizon.

- Service times normalized by a 5-minute reference.

- Binary must-go flag (cannot be deferred to $t+1$).

*Drone-aware attributes*:

1. Drone-eligibility indicator and payload weight (normalized by drone payload limit).

2. Endurance slack: maximum feasible sortie budget for $r$, $\tau_r^{\max}/E$, where $E$ is endurance; 0 if infeasible.

3. Line-of-sight (LoS)/road detour ratio, $\min_k(d_{ik}+d_{kj})/\|i-j\|$, log-scaled and then $z$-normalized.

4. Local opportunity: 5-NN density of drone-eligible neighbors within a fixed radius, log-scaled.

5. Window-slack flags: three binary indicators (tight/mid/loose) derived from normalized slack.

*Normalization.*

- All continuous scalar features are standardized via $z$-score using training-set statistics:

$$\tilde{x} = (x - \mu_x)/\sigma_x.$$

- Coordinates are additionally clipped to the bounding box of each metro area and then mapped to $[-1, 1]$.

- Binary and one-hot features are left as $0/1$.

*Changes in manuscript.*

- New equation (S1) and explanatory bullets added to Sec. 3.1 ("State $s_t$") and Appendix C.1.

- Substantially trim Appendix C.1 and keep only state definitions.

- Clear mapping between "global features" and "epoch instance" added to Appendix C.1.

- Explicit normalization paragraph added to Appendix C.1.

*4. Figure 3 – penalty landscape*

*Reviewer comment.* Figure 3: Could the authors provide quantitative evidence (e.g., learning curves, variance reduction) to justify the claim that the dense penalty improves training stability? Otherwise, consider removing or replacing Figure 3 with empirical results.

*Response.* We appreciate this criticism. In the revision we:

1. Re-label Figure 3 explicitly as a schematic and keep it in Appendix C.2 as part of the penalty description. Its role is now purely explanatory: to illustrate the shape of Eq. (R2).

2. Remove the unsupported comparative claim "proved more stable than a flat FAIL reward" from Sec. 4.1 and Appendix C.2.3. Instead, we refer to the penalty robustness ablation already included in Appendix D.4 (Table 9), which shows that varying the penalty scale $\beta$ leaves final costs essentially unchanged while modestly reducing invalid episodes.

*Changes in manuscript.*

- Figure 3 caption revised to emphasize its schematic nature.

- Main text now references Table 9 in Appendix D.4 when discussing penalty robustness.

- Comparative "proved more stable" sentence removed.

*5. Static drone-truck references*

*Reviewer comment.* Static DAPDP references: The static MILP model in Appendix A is standard in prior literature. Which works did the authors build upon or modify? Please cite them.

*Response.* We now explicitly acknowledge that our static model builds on prior truck–drone formulations. At the start of Appendix A we add:

*"Our static formulation follows the standard truck-and-drone sidekick models introduced by Murray & Chu (2015) for the Flying Sidekick TSP and subsequent TSP-with-drone work (e.g., Agatz et al., 2018), and extends them to multiple trucks with paired pickup–delivery requests and single-leg sorties."*

We then briefly list what is new in our static model: explicit pickup–delivery pairing, one-leg-by-drone constraints (Eqs. (3)–(6)), and the integration with the dynamic dispatch MDP.

*Changes in manuscript.*

- New paragraph at the top of Appendix A referencing Murray & Chu (2015) and Agatz et al. (2018).

*6. Appendix quality and structure*

*Reviewer comment.* Appendix structure: Will the authors reorganize Appendix C to reduce redundancy and clarify environment vs. learning-level definitions?

*Response.* We have streamlined Appendix C and fixed the editorial issues:

1. **Reorganised sections.**

   - **C.1 "MDP and environment"**: Now only contains additional details for state $s_t$, with a single diagram (current Fig. 2).
   - **C.2 "Environment implementation (validation, sampling, penalties)"**: contains solution validation, request generation, and the invalid-solution penalty, including the (now schematic) penalty figure.
   - **C.3 "Learning algorithm: Deep Q-learning"**: consolidates what are currently C.3, C.4, and C.5. We remove the micro-subsections C.3.1–C.3.5 and fold their content into a concise narrative about function approximation, temporal-difference learning, and off-policy Q-learning.

2. **Typos and redundancy.**

   - "Equation equation" is corrected to "Equation".
   - Duplicated explanations of Q-learning in C.3 and C.4 are merged into a single subsection.
   - Text that reads like a basic RL tutorial (e.g., long introductions to tabular Q-learning) is removed, keeping only material that justifies our design choices for dynamic DAPDP.

3. **Environment vs. learning separation.**

   - Environment-level definitions (e.g., Eqs. (18)–(23)) now appear only in C.1–C.2 and are referenced from the main text where necessary.
   - Learning-level updates (Eqs. (25)–(28) and the per-request loss) are grouped in C.3 and Sec. 3.2.2.

*Changes in manuscript.*

- Re-titled and merged Appendix C subsections as described above.

- Corrected typos and removed "Equation equation" repetition.

- Removed redundant text that did not add technical content.

*7. Writing and style / AlphaGo reference*

*Reviewer comment.* AlphaGo reference: Why is AlphaGo mentioned? If used as an analogy, could the authors connect it more explicitly to the reinforcement learning discussion?

Writing and style inconsistencies:

• Reinforcement learning is written in full in Section 3.1 but later abbreviated as reinforcement learning (RL) in Appendix C.5 — inconsistent usage.

• Section C.5.1 introduces AlphaGo without prior context or explanation. The reference appears abrupt and unnecessary.

• Several sentences require stylistic polishing for academic tone and precision.

• Section 2 ('Positioning and Novelty') is overly dense and complex, which makes it difficult to follow. Consider simplifying the language and structure to improve readability.

• The conclusion section should be polished for consistency in tone and tense.

*Response.*

1. **RL / reinforcement learning.** We now introduce the term as "reinforcement learning (RL)" at its first occurrence only.

2. **AlphaGo reference.** We remove the reference to AlphaGo. The reference was meant to serve only as a familiar example of deep value-based RL in complex domains, but is not central to our narrative.

3. **Positioning and novelty (Introduction).** We have rewritten the "Positioning and novelty" paragraph into shorter sentences to improve readability.

4. **Conclusion.** The conclusion is rephrased for stylistic consistency and concision, focusing on: (i) what was done, (ii) empirical performance, and (iii) limitations and future avenues.

*Changes in manuscript.*

- RL terminology made consistent across sections.

- Appendix C.5 AlphaGo paragraph removed.

- Introduction "Positioning and novelty" paragraph simplified.

- Conclusion rewritten for clarity and consistency.