# OpenReview forum: "Dynamic Drone-Assisted Pickup and Delivery Routing"
_ICLR.cc/2026/Conference — Submitted to ICLR 2026_

### Official Review · Reviewer_AYVe · 2025-10-22

**Soundness:** 3
**Presentation:** 2
**Contribution:** 2
**Rating:** 6
**Confidence:** 5

**Summary:**

This paper proposes a Deep Q-Network (DQN) framework to address the Dynamic Drone-Assisted Pickup and Delivery Problem (DAPDP), where new paired orders arrive throughout the day. The system decides dynamically which requests to dispatch, coordinating a fleet of trucks and onboard drones. The authors model the problem as an MDP and train a value-based agent to select subsets of requests for dispatch. Experimental results on large-scale, real-world-inspired datasets show better performance over heuristic and RL baselines.

The problem setting is relevant and timely, and the combination of DRL with combinatorial routing optimization is technically interesting. However, despite numerical improvements, the presentation, clarity, and scientific rigor of the paper are significantly undermined by weak methodological exposition, poor appendix structure, and lack of qualitative visualization.

**Strengths:**

1.	Relevant and timely topic – The paper tackles a timely and complex problem at the intersection of dynamic logistics and heterogeneous fleet coordination (trucks and drones). The dynamic, rolling-horizon setting with stochastic order arrivals is highly relevant to real-world last-mile delivery.
2.	Effective Co-Adaptation: A key insight is that the DRL agent learns to co-adapt with the static sub-solver, selecting request subsets that are easier to solve within the time budget, which is a non-trivial and valuable learned behavior.
3.	Empirical results – Demonstrated measurable cost reduction across realistic datasets, achieving near-oracle performance under certain settings.
4.	Comprehensive ablation studies – Includes sensitivity analysis on learning rate, discount factor, and solver time limits, providing valuable insights into training behavior.

**Weaknesses:**

1. Lack of qualitative visualization
Despite focusing on spatial routing, the paper contains no figures showing truck and drone trajectories, launch/rejoin points, or deferral maps. This omission makes it impossible to assess whether the learned policy demonstrates interpretable or practically meaningful behavior. The results remain purely numerical and fail to provide intuition about what the agent has actually learned.
2. Figure 3 – Conceptually unclear and unsupported
The explanation of Figure 3 (penalty landscape) is inadequate and purely illustrative. The figure provides no empirical content (no learning curves, convergence plots, or variance analysis). The authors claim the “dense signal proved more stable than a flat FAIL reward” but present no numerical evidence. A more useful figure would compare training stability or performance with vs. without this penalty structure.
3. Appendix quality and structure
•	The appendix contains numerous editorial and structural problems:
o	Self-referential typos such as “Equation equation 25.”
o	Over-fragmented subsections (e.g., C.3.1, C.3.2) that add little substance.
o	Redundant or verbose content that reads like documentation rather than a formal academic supplement.
4. State definition inconsistencies
•	The state representation is described twice — in Section 3.1 (compact features for DQN input) and Appendix C.1 (environment-level details) — but with inconsistent terminology and no mapping between them.
•	The paper never defines the state formally (e.g., st=(⋯ )s_t = (\cdots)st=(⋯)), nor specifies which variables are normalized or aggregated.
•	This ambiguity compromises reproducibility and obscures the MDP’s structure.
5. Incomplete reward specification
•	The reward is defined in pieces across Section 3.1 and Appendix C.2.3, without a single unified formula.
•	The interaction between the base reward and penalty terms is unclear.
•	Since the reward function defines the MDP, this version undermines theoretical completeness and reproducibility.
6. Unreferenced static model
•	Appendix A reproduces the standard static DAPDP without citing foundational sources.
•	Acknowledging prior work is necessary for academic integrity and to clarify what is new in this formulation.
7. Writing and style inconsistencies
•	Reinforcement learning is written in full in Section 3.1 but later abbreviated as reinforcement learning (RL) in Appendix C.5 — inconsistent usage.
•	Section C.5.1 introduces AlphaGo without prior context or explanation. The reference appears abrupt and unnecessary.
•	Several sentences require stylistic polishing for academic tone and precision.
•	Section 2 ('Positioning and Novelty') is overly dense and complex, which makes it difficult to follow. Consider simplifying the language and structure to improve readability.
•	The conclusion section should be polished for consistency in tone and tense.

**Questions:**

Visualization and interpretability
	Can the authors include visualizations (maps or trajectory diagrams) showing the spatial behavior of their learned policy?
	For example, how do drone launch points and truck paths differ between the DQN policy and baselines?

	Reward formulation
	What is the exact unified mathematical expression for the reward function used in training, including penalty terms?
	How is the penalty integrated into the Q-learning target update?

	State representation
	Please provide a formal definition of the MDP state space s_t and observation mapping.
	Which features are normalized or transformed before being input to the neural network?

	Figure 3
	Could the authors provide quantitative evidence (e.g., learning curves, variance reduction) to justify the claim that the dense penalty improves training stability?
	Otherwise, consider removing or replacing Figure 3 with empirical results.

	Static DAPDP references
	The static MILP model in Appendix A is standard in prior literature. Which works did the authors build upon or modify? Please cite them.

	Appendix structure
	Will the authors reorganize Appendix C to reduce redundancy and clarify environment vs. learning-level definitions?

	AlphaGo reference
	Why is AlphaGo mentioned? If used as an analogy, could the authors connect it more explicitly to the reinforcement learning discussion?

---

> ### Author Response · Authors · 2025-11-21
>
> We thank the reviewer for commending our work and for taking the time to provide detailed recommendations that have helped us improve the rigor of our paper. We attach a detailed “Response to Comments” file as supplementary material where you can find our detailed response to weaknesses and questions under “Reviewer #5: AYVe”. Below is a summary only.
>
> 1. Visualization and interpretability.
>
> We explain why deferral heat maps provide better visual insight for our scale. These deferral heat maps are now emphasized and referenced in the main text, to illustrate how the learned policy consolidates drone‑eligible work and avoids costly truck detours. Please see the attached detailed response and revised Section 4.4 and Appendix D.1 for discussion.
>
> 2. Reward formulation.
>
> We now provide a single unified description of the reward function, explicitly showing how the base cost and structured penalty terms combine and how the resulting reward is used in the per‑request Q‑learning target. Full notation and the updated text in Section 3.1 and Appendix C.2.3 are described in the attached response.
>
> 3. State representation.
>
> We formally define the MDP state as global features plus per‑request vectors, give a precise mapping from the environment to the DQN input, and specify which features are normalized or transformed prior to training. The attached response details these definitions and cross‑references between Section 3.1 and Appendix C.1.
>
> 4. Figure 3 (penalty landscape).
>
> We have re‑positioned the penalty landscape as a clearly labeled schematic in the appendix, removed the unsupported stability claim, and instead refer to quantitative ablations on penalty scaling that are already reported. The attached response explains these changes and the rationale.
>
> 5. Static DAPDP references.
> We now explicitly acknowledge that our static model builds on standard truck–drone formulations and cite Murray & Chu (2015) and Agatz et al. (2018), clarifying what is adopted versus extended in our formulation. Further details and exact wording added to Appendix A are provided in the attached response.
>
> 6. Appendix structure.
>
> We have reorganized Appendix C to clearly separate environment‑level definitions from learning‑level details, removed redundant micro‑subsections, and corrected editorial issues. The attached response describes the new structure and how it improves clarity and reproducibility.
>
> 7. AlphaGo reference and writing consistency.
>
> We remove the AlphaGo reference from our discussion of deep value‑based RL, standardize “reinforcement learning (RL)” terminology throughout, and simplify the “Positioning and novelty” and conclusion sections for a more consistent academic style. The attached response lists the specific wording changes made in the main text and appendix.

---

> > ### Comment · Reviewer_AYVe · 2025-11-24
> >
> > thanks for these clarification, I maintain my positive rating

---

### Official Review · Reviewer_fXco · 2025-10-23

**Soundness:** 2
**Presentation:** 2
**Contribution:** 2
**Rating:** 4
**Confidence:** 3

**Summary:**

The paper analyzed the dispatching problem for the last-mile delivery using drone-truck integration. A MDP is formulated and experiment is conducted on a simulated data-set.

**Strengths:**

+ The problem is novel and interesting. It’s also more complicated compared to existing last-mile delivery problem.
+ The formulation of the problem is relatively complete and clear.

**Weaknesses:**

- The major contribution of the paper seems to be in the formulation, not sure it meets the expectation for a ICLR paper.
- The assumption that orders come randomly is not realistic. Last-mile orders, especially for food and grocery delivery, has strong spatial and temporal patterns.
- Grammar. “the agent must time dispatch decisions” on page 2.

**Questions:**

See weaknesses.

---

> ### Author Response · Authors · 2025-11-21
>
> We thank the reviewer for taking the time to review our paper and for commending our work as novel and interesting. Please find our responses to the weaknesses highlighted below:
>
> Weakness 1
>
> We agree the static formulation alone would not warrant ICLR. Our contribution is not the static model but a decision‑focused RL policy that (i) learns subset selection (dispatch vs. defer per request) that is drone‑aware (endurance slack, LoS/road detour ratio, local 5‑NN density) and (ii) co‑adapts with a transparent ALNS sub‑solver under tight time windows and single‑leg drone sorties. This decouples “what to dispatch now” from “how to route,” yielding a controller that discovers when to consolidate drone‑suitable work and when to defer isolated requests. We foreground this in the revised “Positioning and novelty” paragraph and the analysis section (deferral heatmaps, Q‑slices). Ablations show that removing drone‑specific observables degrades performance (+2.2% to +3.4%), and disabling the drone entirely raises cost by +7–8%, demonstrating learned, UAV‑specific behavior rather than a generic VRP heuristic (Sec. 3.1, 4.4; App. D.1, D.6). Table 2 further shows our policy outperforms strong non‑learning (MSA/SAA, waiting) and PPO baselines at matched compute.
>
> Weakness 2
>
> Our dynamic instances are not sampled from a synthetic uniform field. They are revealed by sampling without replacement from 300 real delivery days (two metro areas), preserving the empirical spatial layout and time‑window structure of actual operations; we overlay drone feasibility on those logs (Sec. 4.1 “Data provenance”; App. C.2.2). The state fed to DQN includes hour‑of‑day (sin/cos), time‑window slack flags, and local spatial density so the policy can exploit temporal rhythms and clustering when deciding to defer or dispatch (Sec. 3.1; App. C.1). To stress realism and transfer, App. D.8 evaluates generalization from City‑A to a held‑out City‑B with different road density/tightness and shows the mixed‑curriculum model retains near‑oracle performance on A while doubling generalization on B relative to training on A only. This is evidence the learned rule tracks real structure rather than overfitting a sampling artifact. We also release a synthetic generator that matches the marginals of the confidential data for full reproducibility (App. E.3). (Sec. 4.1; App. C.1, C.2.2, D.8, E.3.)
>
>
> Weakness 3
>
> This whole paragraph has been re-written for clarity.

---

### Official Review · Reviewer_vyDR · 2025-10-28

**Soundness:** 1
**Presentation:** 1
**Contribution:** 1
**Rating:** 2
**Confidence:** 5

**Summary:**

This paper studies the dynamic drone-assisted pickup and delivery problem (DAPDP), where a fleet of trucks equipped with drones serves dynamically arriving paired pickup–delivery requests under time-window, capacity, and endurance constraints. The authors propose a deep Q-learning (DQN)–based approach to decide which new orders to dispatch and how to coordinate drone sorties. A paired ALNS (adaptive large neighborhood search) sub-solver is used for route construction. Experiments on a “real-world-inspired” dataset with up to 200 customers reportedly show that the method outperforms greedy, random, and PPO baselines and achieves performance close to a clairvoyant oracle.

**Strengths:**

1. The topic — dynamic pickup-and-delivery with drones — is practically relevant and fits the growing interest in combining reinforcement learning with combinatorial logistics optimization.
2. The empirical results, if valid, suggest that learning-based decision rules might yield efficiency gains in dynamic dispatching environments.
3. The integration of an RL agent with an optimization-based sub-solver is conceptually interesting and aligns with decision-focused learning paradigms.

**Weaknesses:**

1. Lack of technical novelty.
The proposed approach relies on a standard deep Q-learning framework with minor heuristic adjustments. There is no clear algorithmic or theoretical innovation beyond existing DQN formulations or hybrid RL–metaheuristic approaches.
2. Insufficient methodological rigor.
The MDP formulation and environment definition are vague and incomplete. Key components — such as state representation, transition dynamics, and reward specification — are not defined rigorously, making it difficult to assess reproducibility or correctness.
3. Unjustified design choices.
The integration of ALNS into the framework is insufficiently explained. It remains unclear why ALNS is chosen, how its neighborhoods are designed, what hyperparameters or termination criteria are used, or how it interacts with the learning policy. Without such detail, the claimed near-optimal performance is not verifiable.
4. Unrealistic or underspecified experimental setup.
Several assumptions are overly restrictive or inconsistent with realistic dynamic pickup-and-delivery settings — most notably the “all requests must be served” constraint. Furthermore, the dataset description lacks transparency: the origin and realism of the 200-customer scenario are unclear, and there is no evidence that the environment captures meaningful stochasticity or dynamism.
5. Questionable baselines and results.
The claim that the proposed DQN approach achieves performance within 1% of a clairvoyant baseline is implausibly strong and not supported by proper justification. Details on the oracle baseline, its computational budget, and its use of future information are missing. No ablation or sensitivity analysis is provided to understand why such high performance is achieved.
6. Limited generalization and insight.
The study provides no conceptual or methodological insights that would generalize beyond this specific problem. As a result, the contribution is primarily empirical and lacks depth expected for ICLR.

**Questions:**

1. What are the hyperparameters and termination criteria for the ALNS sub-solver, and how is its performance benchmarked?
2. How is the clairvoyant (oracle) baseline implemented, and what future information does it assume access to?
3. Could the authors provide details about the dataset generation process and justify why 200 customers represent a realistic operational scale?
4. Have the authors tested generalization across different levels of dynamism or uncertainty in request arrivals?

---

> ### Author Response · Authors · 2025-11-21
>
> We thank the reviewer for the detailed and critical feedback. Many of the concerns center on methodological clarity and the visibility of details that were previously largely in the appendices. In the revised version we have moved key definitions into the main text, and highlighted the description of the sub‑solver and oracle, and generalization and ablation results.
> Please see our detailed “Response to Comments” file attached as supplementary material under section “Reviewer #3: vyDR” where we address each weakness and question in turn.
>
> In our response, We clarify the MDP, state, and reward definition in the main text; clarify hyperparameters, neighborhoods, and benchmarking for the ALNS sub-solver and oracle; and foreground scenario-planning, waiting, and PPO baselines as well as generalization experiments across cities, spatial distributions, and fleet caps. These changes, together with a more explicit novelty discussion and foregrounded managerial insights, address the reviewer’s concerns about rigor, realism, and broader contribution.

---

### Official Review · Reviewer_ecQu · 2025-10-29

**Soundness:** 2
**Presentation:** 2
**Contribution:** 2
**Rating:** 4
**Confidence:** 3

**Summary:**

This work investigates the dynamic drone-assisted pickup and delivery problem and proposes a deep reinforcement learning (DRL) approach based on deep Q-learning, to decide dynamically which newly arrived orders to dispatch and how to integrate drone sorties effectively.  However, it is not clear how to use the DQN to solve the challenges in the dynamic drone-assisted pickup and delivery problem, such as the dynamic orders and the cooperation of trucks and drones. Additionally, this work lacks the experiments of comparing with the baselines in the truck-drone delivery.

**Strengths:**

1. This work models the dynamic drone-truck collaborative delivery problem with time windows aligns with emerging research topics in the current logistics industry, boasting high application value and cutting-edge relevance.
2. This paper applies the standard DQN to learn dispatching decisions, while the complex routing problem is optimized by a dedicated traditional optimizer ALNS. This is a highly practical and effective choice, and the experimental results also demonstrate the superiority of this method.

**Weaknesses:**

1. The generalization experiments explore scenarios with different urban distributions. What is the model's generalization performance when the request scale of test instances (e.g., 300 requests) is much larger than that during training (100-200 requests)?
2. The experimental results do not include a comparison of inference time. Adding this comparison would enable a better understanding and evaluation of the method, as it is important to know whether ALNS optimization is time-consuming.
3. The addition of shaped rewards is mentioned in Section 4.5.1, but no specific details are provided.
4. Regarding the reward_for_decision in Section 3.2.2, the reward for one step is evenly distributed among each node, which seems unreasonable. For example, if A, B, and C are selected—where A is far from B and C, while B and C are close to each other—this single action returns a large negative reward. A, B, and C receive the same penalty, but in this case, the priority should be to avoid selecting A as much as possible.

**Questions:**

1. It is not clear how to use the DQN to solve the challenges in the dynamic drone-assisted pickup and delivery problem, such as the dynamic orders and the cooperation of trucks and drones.

2. Could you compare this method with the state-of-the-art method in the truck-drone delivery problems? This work lacks the experiments of comparing with the baselines in the truck-drone delivery.

---

> ### Author Response · Authors · 2025-11-21
>
> We thank the reviewer for their thoughtful comments and for recognizing the practical relevance and effectiveness of combining DQN with a dedicated ALNS sub-solver.
> We address each weakness and question in detail in the “Response to Comments” file we attach as supplementary material. Please find the detailed response under “Reviewer #2: ecQu” in that document. Below is a summary only.
>
> Weaknesses:
> 1. Generalization to larger request scales.  We clarify that 300-or more real requests testing is part of future work and that the DQN operates per-request while the ALNS sub-solver handles the combinatorial routing, so scalability is driven mainly by the static solver; we also highlight the generalization section (Appendix D.8) that discusses how the learned policy transfers across cities and fleet sizes and added an explicit note on scaling to larger request sets (see attached detailed response).
>
> 2. Missing inference-time comparison. We now report wall-clock inference times and solver runtimes explicitly in Section 4 and Appendix E.1, showing that a full epoch (network forward pass plus ALNS) takes only a few seconds, and we discuss the time–quality trade-off across different ALNS time limits (Figure 1 and Table 3).
>
> 3.  Lack of detail on shaped rewards. The reward specification has been consolidated and clarified, and Section 4.5.1 includes an ablation showing that shaped rewards both accelerate convergence (≈30% fewer training steps) and improve final cost by 6–8% relative to using only the base cost signal.
>
> 4. Per-request reward allocation. We clarify in Section 3.2.2 and Appendix C.1 that the per-request reward shares a global cost signal across dispatched requests, and we explain how, through repeated experience and replay, the agent learns to favor dispatch patterns that exclude systematically harmful requests (like isolated, far-away nodes); we also note that more sophisticated credit-assignment schemes are an interesting extension, but the current design already yields strong empirical performance.
>
>
> Questions:
> 1. We clarify that the DQN learns a per-request dispatch/defer policy using rich drone-aware features, while a static truck–drone ALNS sub-solver handles routing and launch–rendezvous decisions; this two-level architecture lets the agent handle dynamic arrivals and vehicle cooperation by learning which subsets are cheap and feasible to route. See attached response and Appendix C.1 for details.
>
> 2. We now explicitly connect our static model to standard truck–drone formulations and argue that we compare against strong dynamic baselines (MSA/SAA, Waiting/Relocation, PPO) that all use the same truck–drone ALNS sub-solver, showing 2.6–6.0% cost improvements and a 1.2% gap to a clairvoyant Oracle. See attached response and Sections 4.1–4.3, B.3, and Appendix A/D.2.

---

### Official Review · Reviewer_k2s8 · 2025-11-10

**Soundness:** 2
**Presentation:** 2
**Contribution:** 2
**Rating:** 2
**Confidence:** 4

**Summary:**

This work proposes a deep Q learning approach for a dynamic drone-assisted pickup and delivery routing problem. The authors attempt to address some of the main challenges of this last-mile delivery problem, such as dynamic requests,  coordination of the ground vehicle and the drone, etc. The static version of the problem is formulated as a Markov Decision Process, with the state space, the action space being a binary variable (dispatch or defer), and the reward being reflective of the total distance as cost, along with the constraint violation penalty after solving the subproblem. The approach was compared against 7 other baseline methods. Parametric study, ablation studies, and hyperparameter sensitivity studies were also performed.

**Strengths:**

The problem addressed in this manuscript is both highly relevant and inherently challenging within the context of Last Mile Delivery (LMD) — a domain that plays a critical role in modern logistics and e-commerce operations. Efficiently optimizing routes and resource allocation in this stage directly impacts delivery speed, operational cost, and customer satisfaction, making it a central focus of contemporary research in operations research and transportation systems. The authors have effectively captured the complexity of this problem through a well-formulated optimization framework that accurately reflects the real-world constraints and dynamic nature of LMD scenarios. The mathematical formulation is rigorous and thoughtfully constructed, providing clear insights into the trade-offs and decision variables involved. Furthermore, the detailed problem description enhances the manuscript’s clarity and accessibility, allowing readers to fully appreciate the technical depth and practical significance of the proposed approach. Overall, the problem formulation and presentation demonstrate a strong alignment with real-world logistics challenges and contribute meaningfully to advancing optimization methodologies for last-mile delivery systems. The problem considered in this manuscript is a very relevant and difficult problem in Last Mile Delivery. The optimization formulation of the problem, along with the detailed description, is well appreciated.

**Weaknesses:**

1. Even though the practicality of this work is unquestionable, and the authors have formulated the static version of the problem as an MDP, there is no novelty in the methodology used (simple deep Q learning). Even though it can be argued that there is no need for a more novel, sophisticated approach, given the focus of ICLR, I feel  ICLR might not be the right platform for showcasing this work, and is more apt for an optimization-related platform.

2. The writing style of the manuscript can also be significantly improved. At present, the writing style resembles more of a project report than an academic paper. For example, the meaning of many terms in the state space is not clear. I encourage the authors to improve the clarity of writing there, and also in many other parts of the manuscript.

3. The current choice of baseline appears relatively weak and limits the strength of the comparative analysis. To provide a more convincing evaluation, the study would benefit from incorporating stronger and more representative baselines. In particular, since the Oracle method assumes complete knowledge of the system — an unrealistic assumption in practical settings — it would be more appropriate to replace or complement it with a Mixed Integer Linear Programming (MILP) formulation. MILP-based methods serve as a more rigorous and interpretable benchmark, offering a well-established optimization standard against which the proposed approach’s performance can be meaningfully compared. Including such a baseline would not only strengthen the empirical validation but also highlight the practical advantages and limitations of the proposed method under realistic conditions.

**Questions:**

1. What is 60s ALNS?

2. How is the sub-problem solved during every decision-making step?

3. The experimental implementation details are not provided. Can the authors please provide how the environment is implemented using the dataset? For example, the programming platform, or other simulation environment, if any.

---

> ### Author Response · Authors · 2025-11-21
>
> We thank the reviewer for recognizing the relevance and difficulty of the dynamic drone‑assisted pickup and delivery problem (DAPDP). While we do employ a standard DQN, our contributions lie in the DAPDP-specific RL formulation. That is, drone-aware state features, per-request Q-learning coupled with a static paired truck–drone solver, and structured penalties, which together yield a dispatch policy that achieves a 1.2% oracle gap and 6.6–37.2% improvements over strong optimization and RL baselines. We have substantially revised the exposition (state definition, terminology, environment details) and clarified the role of the oracle and MILP formulation, and we hope these changes address the reviewer’s concerns about novelty, clarity, and empirical rigor.
> We include much more detail in the “Response to Comments” file we attach as supplementary material. Please find the detailed response under “Reviewer #1: k2s8” in that document. Below is a summary only of our response.
>
> Weaknesses:
> 1. While the RL backbone is a standard DQN, the novelty lies in the DAPDP-specific RL formulation: drone-aware state features, per-request Q-learning for subset dispatch, and structured penalties co-adapted with a paired static sub-solver, yielding a learned policy that achieves a 1.2% oracle gap and improves over strong optimization and RL baselines. We argue this is of direct interest to the ICLR community working on neural combinatorial optimization and dynamic routing.
>
> 2. We revised Section 3.1 and Appendix C.1 to formally define the MDP state and clarify each feature (e.g., “drone-eligibility indicator,” “line-of-sight (LoS) / road detour ratio”), making the state space and its normalization unambiguous.
>
> 3. We already compare against several strong optimization and RL baselines at matched compute, and we now clarify that the oracle is a clairvoyant lower bound; while we provide a static MILP formulation, exact MILP solves are intractable beyond very small instances.
>
> Questions:
> 1. We now spell out that “60s ALNS” means running the same ALNS sub-solver as in Section 4.1 with a 60‑second time limit, and we cross-reference the detailed description in Appendix E.2.
>
> 2. At each epoch the agent’s dispatch subset defines a static DAPDP, which we solve with the ALNS-lite sub-solver (Algorithm 2) under a fixed per-epoch time limit, and the resulting route cost and feasibility determine the agent’s reward.
>
> 3. We added a short description at the start of Section 4 and explicit pointers to Appendix C.2 and E.1, which describe the Python/PyTorch environment, request generation, and hardware/software setup in detail.

---

### Meta-Review · Area_Chair_VTBM · 2026-01-05

**Summary:**

Reviewers' concerns are mainly around the following points:

W1. Insufficient technical novelty

W2. Writing does not reach the standard of ICLR

W3. Baselines are weak

W4. Lacking inference time comparison

W5. Lacking details of reward shaping

W6. Unjustified design choices

W7. Unrealistic or underspecified experimental setup

**Reviewer Concerns:**

The rebuttal addressed the methodology clarification related issues (e.g., W4, W5, W6). However, concerns regarding the insufficient novelty (W1), problematic experiments (W3, W6, W7) and writing (W2) largely remains.

**Reviewer Scores:**

Only the positive reviewer AYVe participated the discussion, and decided to maintain his/her evaluation. For the other reviewers, I feel that authors' rebuttal is not sufficient to turn their evaluation into positive.

---

### Decision · Program_Chairs · 2026-01-26

Reject